# Exploring the repository of de novo-designed bifunctional antimicrobial peptides through deep learning

Ruihan Dong[1,2†], Rongrong Liu[3†], Ziyu Liu[3†], Yangang Liu[4†], Gaomei Zhao[5], Honglei Li[6], Shiyuan Hou[3], Xiaohan Ma[3], Huarui Kang[3], Jing Liu[3], Fei Guo[7], Ping Zhao[4], Junping Wang[5], Cheng Wang[5*], Xingan Wu[3*], Sheng Ye[1*], Cheng Zhu[1*]

[1]Frontiers Science Center for Synthetic Biology (Ministry of Education), Tianjin Key Laboratory of Function and Application of Biological Macromolecular Structures, School of Life Sciences, Faculty of Medicine, Tianjin University, Tianjin, China; [2]Center for Quantitative Biology, Academy for Advanced Interdisciplinary Studies, Peking University, Beijing, China; [3]Department of Microbiology, School of Basic Medicine, Fourth Military Medical University, Shaanxi, China; [4]Department of Microbiology, Second Military Medical University, Shanghai, China; [5]State Key Laboratory of Trauma and Chemical Poisoning, Institute of Combined Injury of PLA, College of Preventive Medicine, Third Military Medical University (Army Medical University), Chongqing, China; [6]Tianjin Cancer Hospital Airport Hospital, Tianjin, China; [7]School of Computer Science and Engineering, Central South University, Changsha, China

*For correspondence:
wangctmmu@126.com (CW);
wuxingan@fmmu.edu.cn (XW);
sye@tju.edu.cn (SY);
cheng_zhu@tju.edu.cn (CZ)

†These authors contributed equally to this work

## eLife Assessment

This study presents a **useful** pipeline for de novo design of antimicrobial peptides active both against bacteria and viruses. The method is based on deep learning, using a GAN generator and a regression tasked to predict antimicrobial activity. The experimental evidence supporting the conclusions is **solid**, with 24 validated peptides, although some additional justifications of the computational strategy would be a plus. This work will be of interest to the community working on machine learning for biomedical applications and specifically on antimicrobial peptides.

**Abstract** Antimicrobial peptides (AMPs) are attractive candidates to combat antibiotic resistance for their capability to target biomembranes and restrict a wide range of pathogens. It is a daunting challenge to discover novel AMPs due to their sparse distributions in a vast peptide universe, especially for peptides that demonstrate potencies for both bacterial membranes and viral envelopes. Here, we establish a de novo AMP design framework by bridging a deep generative module and a graph-encoding activity regressor. The generative module learns hidden 'grammars' of AMP features and produces candidates sequentially pass antimicrobial predictor and antiviral classifiers. We discovered 16 bifunctional AMPs and experimentally validated their abilities to inhibit a spectrum of pathogens in vitro and in animal models. Notably, P076 is a highly potent bactericide with the minimal inhibitory concentration of 0.21 µM against multidrug-resistant *Acinetobacter baumannii*, while P002 broadly inhibits five enveloped viruses. Our study provides feasible means to uncover the sequences that simultaneously encode antimicrobial and antiviral activities, thus bolstering the function spectra of AMPs to combat a wide range of drug-resistant infections.

## Introduction

In the post-antibiotic era, microbes rapidly evolve to become antimicrobial-resistant under the pressure of antibiotic abuse and through horizontal transfer of resistance genes (*Lewies et al., 2019*). Due to increasing cases of untreatable bacterial infections, the World Health Organization (WHO) has projected a looming health threat of 10 million casualties per year by 2050 (*Murray et al., 2022*). Specifically, the WHO has identified six antimicrobial-resistant pathogens (*Enterococcus faecium*, *Staphylococcus aureus*, *Klebsiella pneumoniae*, *Acinetobacter baumannii*, *Pseudomonas aeruginosa*, and *Enterobacter* species) as ESKAPE. The ESKAPE pathogens have severely limited the treatment options for serious infections and impaired the conditions of immunodeficient patients (*De Oliveira et al., 2020*). In sharp contrast, the majority of clinically available antibiotics were developed more than 30 years ago (*Hutchings et al., 2019*). Hence, there are urgent needs for targeted generation and screening of novel molecules with desired therapeutic properties for the next-generation broad-spectrum antimicrobial agents.

As a naturally evolved defense line of innate immune system, antimicrobial peptides (AMPs) are cationic macromolecules (usually 10–60 amino acids) widely present in all living organisms (*Huan et al., 2020*). AMPs are attractive alternatives to conventional antibiotic treatment because they effectively inhibit a wide range of pathogens, such as the Gram-positive and Gram-negative bacteria, fungi, and viruses (*Huan et al., 2020*). The majority of AMPs featured amphiphilic patterns of amino acid sequences and enriched positive charges (e.g., lysine and arginine residues), which allowed AMPs to disrupt the membrane components and exert antimicrobial effects (*Mahlapuu et al., 2016*). Due to the relative constant compositions of biological membranes as the essential AMP targets, development of resistance to AMPs was a rare event even after 100 passages of lab evolutions (*Spohn et al., 2019*; *Huang et al., 2023*).

Notably, the naturally evolved and rationally designed AMP sequences (~20,000) are housed in several databases (APD [*Wang et al., 2016*], DBAASP [*Pirtskhalava et al., 2021*], DRAMP [*Shi et al., 2022*], etc.) and paired with experimentally validated activities (minimal inhibitory concentration [MIC]), thus facilitating data mining endeavors for more diverse sequences and more effective peptides (*Xu et al., 2021*). Indeed, deep learning models have led to the discovery of highly efficient AMPs (e.g., MIC of 2 μM against *Escherichia coli*). Specifically, predictive models like Deep-AMPEP30 (*Yan et al., 2020*), AMPlify (*Li et al., 2022*), TransImbAMP (*Pang et al., 2022*), and sAMPpred-GAT (*Yan et al., 2023*) utilized deep convolutional, recurrent, attention-based, and graph neural networks to learn and recognize latent representations of meaningful peptides. These models recently enabled comprehensive mining of the human gut microbiome (*Ma et al., 2022*), modern and ancient human proteomes (*Maasch et al., 2023*; *Torres et al., 2022*), or even the entire sequence space of hexapeptides (*Huang et al., 2023*). Generative models like PepVAE (*Dean et al., 2021*), CLaSS (*Das et al., 2021*), and HydrAMP (*Szymczak et al., 2023*) reconstructed the sequences from latent space according to the encoder-decoder architecture of variational autoencoders (VAEs), while PepGAN (*Tucs et al., 2020*) and AMPGAN v2 (*Van Oort et al., 2021*) utilized the generative adversarial network (GAN) to approximate the antimicrobial sequences by deceiving the discriminator.

Despite the recent advances of data-driven AMP classifiers and generators, the exploration of a vast AMPs repertoire is still hindered by several factors: (1) the sparsity of AMPs in the whole peptide space, which demands accurate prediction of MIC values for efficient sampling and filtering of AMP analogues (*Szymczak et al., 2023*; *Pandi et al., 2023*). The current machine learning predictive models usually perform the binary classification task and label the generated sequences as active or not, instead of giving specific activity values. To overcome this shortcoming, a popular pipeline combines independent modules of filtering, classification, and regression, but it faces risks of step-wise bias accumulation (*Huang et al., 2023*). (2) The impact of different phospholipid species on the efficacy of AMPs. Due to the data incompleteness of antimicrobial measurements, a majority of AMP classifiers or MIC predictions were specific to *E. coli* (Gram-negative). Nevertheless, these models were applied to find peptides that inhibit other Gram-positive bacteria (e.g., *S. aureus*), based on the assumption that the shared membrane components are potential targets of AMPs. More importantly, the enveloped virus also possesses a targetable membrane, which closely mimic the phospholipid components of cellular organelles. Even though the AMP database recorded antiviral activities of at least 2000 antiviral peptides (AVPs), whether the current models can readily generate AVPs remain elusive. (3) Lack of rigorous preclinical models. To advance the designed AMPs toward clinical trials,

the experiments should reflect the efficacy of AMPs in tissue samples and require the determination of safe doses of a particular sequence before practical applications. To our knowledge, more than 120 AMPs have been suggested by deep learning models in the past three years, only a fraction of these studies demonstrated animal results.

In the current study, we addressed these shortcomings and presented a new deep learning-based de novo AMP design framework, consisting of a GAN generator and an antimicrobial activity regressor named AMPredictor. The graph convolution network (GCN) of AMPredictor allowed us to effectively train a regression model for the task of MIC predictions. Through in-depth investigations of generated sequences (e.g., antiviral possibilities), we selected three top-ranking sequences (P001, P002, and P076) and evaluated their efficacies against different pathogens (four drug-resistant bacteria and five enveloped virus). Out of the three candidates, we found P076 is a potent bactericide with MIC of 0.21 µM against multidrug-resistant (MDR) bacterium. It also outperformed a clinically relevant antibiotic polymyxin B (PB) in terms of safety and efficacy, as evaluated in mouse models. Another peptide P002 broadly neutralized viruses in cellular infection models. We challenged our computational models to generate bifunctional AMPs, and our results demonstrated that these peptides inhibited a spectrum of pathogens, but with delicate preferences toward bacterial or viral species.

## Results

### Bridging the generator and regressor for de novo AMP design

The overall architecture for designing functional peptides fuses a GAN and an antimicrobial activity predictor (AMPredictor, *Figure 1a*). We aim to explore beyond the known antimicrobial templates or accessible proteomics space. We trained the GAN module with 3280 AMP sequences to capture the pattern of AMPs through few-shot learning. The training goal of GAN is to reach a relative balance between the generator and discriminator (see *Equation 1*). The generator takes random noise as input to produce fake samples, which are fed into the discriminator along with the real data. The discriminator tries to distinguish between the real and fake data, then returns a probability to update the weights of the networks (*Goodfellow et al., 2014*). Here, we employed a low-dimensional vector called amino acid factors (AAFs) to encode the peptide sequences and describe their physicochemical features (*Atchley et al., 2005*). We monitored the generated sequences at different training stages (epoch 1, 100, 200, 400, 800) and visualized the optimization process using two common dimension reduction approaches, uniform manifold approximation and projection (UMAP) and t-distributed stochastic neighbor embedding (t-SNE). We clearly observed the distributions of newly sampled sequences among the existing AMPs after mapping peptide AAF features onto the two-dimensional space (*Figure 1b*). With iterating and updating, the generator learns the feature distribution of actual data and generates diverse sequences.

We checked the novelty and diversity of generated sequences at epoch 1000 by aligning them with our training set. The sequence similarities were evaluated by global alignment using BLOSUM62 matrix. We found the identity of each generated sequence was around 0.35 to the known database (*Figure 1c*), indicating a low level of similarity to the existing AMPs. With distinct sequence compositions, generated peptides still hold similar physiochemical features to real AMPs, as evidenced by six descriptors of aliphatic index, aromaticity, net charge, the ratio of hydrophobic amino acids, instability index, and isoelectric point (*Figure 1d*; *Müller et al., 2017*). The distribution of these descriptors closely resembled the physiochemical features of AMPs in the training set. Hence, adopting residue features as the input of GAN generator can capture the functional properties of actual AMPs.

We then developed an independent module AMPredictor to estimate the MIC values of arbitrary peptide sequences. AMPredictor is a GCN-based regression model, equivalent to the 'score function' in virtual drug screen. Unlike the low-dimensional descriptor adopted by GAN generator, we designed distinct representations of peptides in AMPredictor for model accuracy and orthogonal verification of GAN-generated sequences. We chose graph as the representation of peptides, with their amino acid residues as graph nodes and the predicted contact maps as graph edges. The node features were embeddings from the protein language model ESM (*Rives et al., 2021*). The peptide graphs were then fed into a three-layer GCN to learn the hidden 'grammars' of AMP features. In addition to the sequence-level and structural-level features, we transformed the input amino acid sequences into Morgan fingerprints to encode the chemical features at each residual position. This

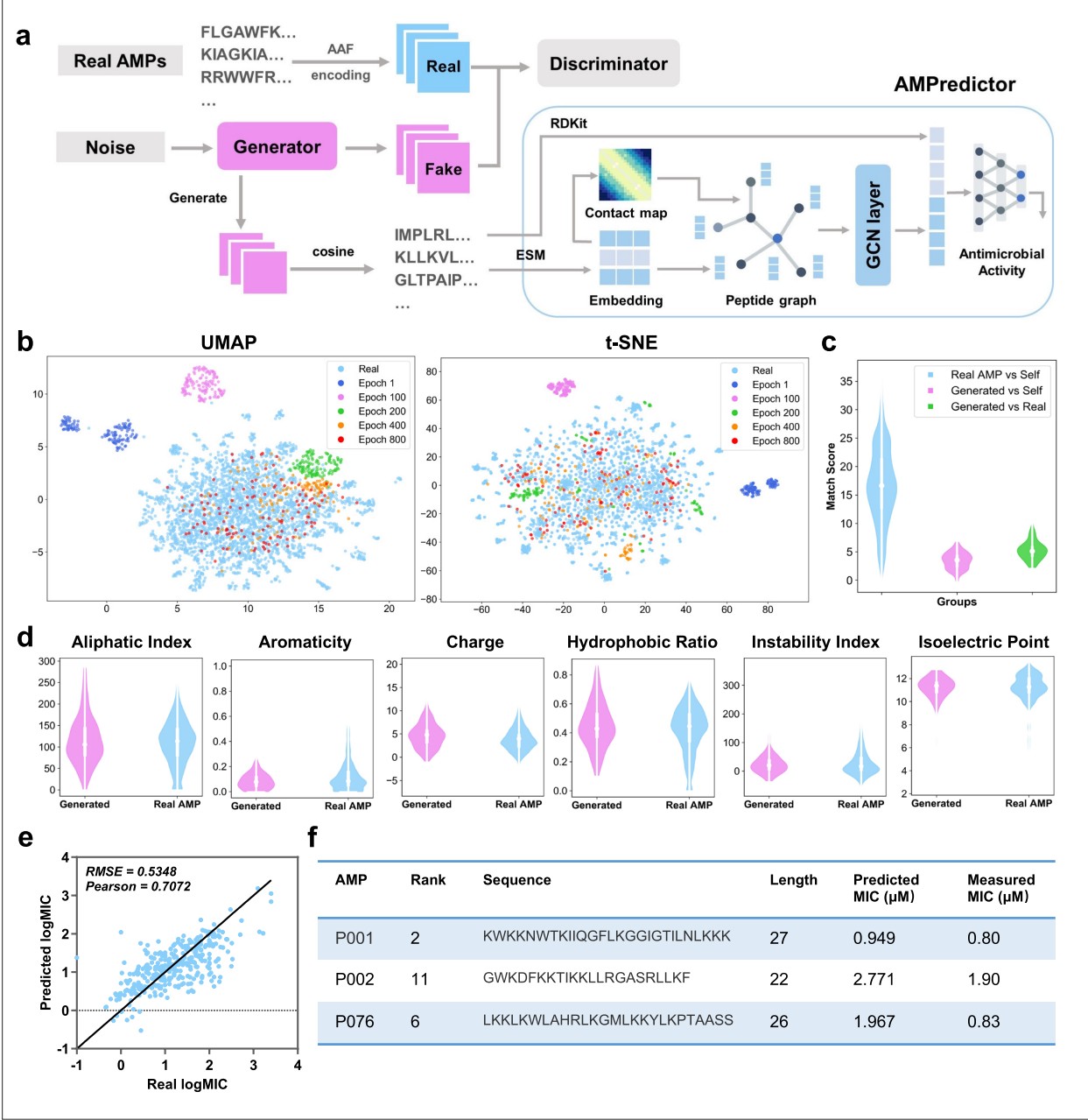

**Figure 1.** Deep learning-based design of novel antimicrobial peptides. (**a**) The overall framework of generative adversarial network (GAN) generator and antimicrobial activity predictor (AMPredictor). (**b**) Visualization of GAN training process via uniform manifold approximation and projection (UMAP) and t-distributed stochastic neighbor embedding (t-SNE). The newly generated sequences fit the distribution of known antimicrobial peptide (AMP) space gradually. (**c**) Global alignment match scores between the generated sequences and itself and AMPs in the training set. (**d**) Six physiochemistry properties of generated peptides and real AMP sets. (**e**) Regression results of AMPredictor model on its test set. (**f**) The sequences of three selected peptides as well as their predicted and experimentally validated minimal inhibitory concentration (MIC) values against *E. coli* ATCC 25922.

The online version of this article includes the following figure supplement(s) for figure 1:

**Figure supplement 1.** Dataset and hyperparameter settings of antimicrobial activity predictor (AMPredictor).

**Figure supplement 2.** Antimicrobial activity regression results on test set of five baseline models.

**Figure supplement 3.** Predicted structures and the helical wheel projection of three selected peptides.

atomic-level information was concatenated with GCN representations before entering a three-layer fully connected networks to regress on MIC datasets (*Figure 1a*). AMPredictor obtained a low root mean squared error (RMSE) of 0.5348 and a high Pearson correlation coefficient (PCC) of 0.7072 on the test set (10% size of the dataset) (*Figure 1b*). The output of AMPredictor is the logMIC value of an AMP sequence.

By integrating the representations of residual-level, sequence-level, and protein structural-level information, AMPredictor obtained the best performance in comparison to its ablating versions (*Supplementary file 1a*). The ablation study indicated that the context embeddings from ESM play vital roles in predicting MICs, and the contact map assisted in improvement despite it being a simplified representation of protein structure. We also compared the performance of AMPredictor with five typical deep neural networks as baselines, including the multilayer perceptron (MLP), convolutional neural network (CNN), gated recurrent unit (GRU), long short-term memory (LSTM), and Transformer. AMPredictor outperforms these methods in terms of accuracy at the task of predicting antimicrobial activity values (scatter plots in *Figure 1—figure supplement 2*). Hence, a well-trained generator could produce a variety of AMP candidates and then AMPredictor ranked the candidates by MIC values. Together with coarse to fine representations of AMP samples, the GAN and AMPredictor facilitated de novo design and in silico evaluation of AMPs.

## Designing bifunctional antimicrobial peptides

Our study was based on the hypothesis that both bacterial membranes and viral envelopes were potential targets of AMPs, and the dual-function of antibacterial and antiviral could be encoded into one peptide sequence. We directly compared the outcome of two different training sets for two versions of the GAN generator: with or without adding AVP sequences, for their efficiencies of designing bifunctional peptides (*Figure 2a*). The generator v1 was trained solely on collected AMP sequences, and generator v2 was trained on the AMP set and known 1788 AVPs from AVPdb. To establish a sequential workflow and filter out AMPs with antiviral capabilities, we then considered assembling several AVP classifiers subsequent to the AMPredictor. As the source data, the AVPdb provided ~2000 sequences with $IC_{50}$ labels (*Qureshi et al., 2015*). However, data curation resulted in only 759 unique AVPs. We determined that the quantitative data of AVPs were insufficient for training another GCN-based regression model. Instead, binary classification should exhibit robustness against noise of a relatively small dataset. Hence, we merely made yes/no decisions using a collective of five published AVP classification tools (AVPpred [*Thakur et al., 2012*], DeepAVP [*Li et al., 2020*], Deep-AVPpred [*Sharma et al., 2022*], ENNAVIA [*Timmons and Hewage, 2021*], and AI4AVP [*Lin et al., 2022*], details in *Supplementary file 1b*). We obtained 1024 new peptides from each generator and kept the sequences with a predicted MIC value of less than 10 μM by AMPredictor. More than 100 peptides passed the ensemble filter of all five AVP classifiers. Interestingly, generator v2 did not feature a higher passing ratio, although it was trained with AVPs. Based on their hemolysis and other physicochemical properties, we selected 12 peptides from each generator for experimental validation. We also checked their novelty using BLAST with the training set or DRAMP database, and all their E-values are greater than 0.1 or no hits found (*Supplementary file 1c*).

All 24 peptides exhibited effective antibacterial capability in inhibiting six different strains involving three MDR species (*Figure 2b*). Several peptides obtained remarkable MIC values of less than 1 μM, such as P026 and P036, indicating the effectiveness of our generator for de novo designing AMPs. Then we used a stricter rule of 'antibacterial' to define bifunctional peptides. If the MIC value of a peptide was less than 10 μM against at least one strain, we described it as antibacterial. We also tested whether these peptides can inhibit Herpes simplex virus 1 (HSV-1) at the concentration of 10 μM. The 'antiviral' feature of a peptide was checked when significant inhibition was observed in immunofluorescence experiments. There were 6 and 7 bifunctional AMPs out of 12 from generators v1 and v2, respectively (*Figure 2c*). Given that we did not screen on more virus species, the success rate of being antiviral is relatively lower than being antibacterial. This result indicates the capability of our design method for bifunctional AMPs. Through the comparison between generators v1 and v2, we demonstrated that the antiviral feature was 'naturally' encoded in AMP sequences.

We then paired small-scale design with more comprehensive experimental characterization, using four drug-resistant bacteria, five enveloped viruses, and animal infection models, focusing on generator v1. We acquired 104 novel peptides via the de novo design framework. Among them, 42 and

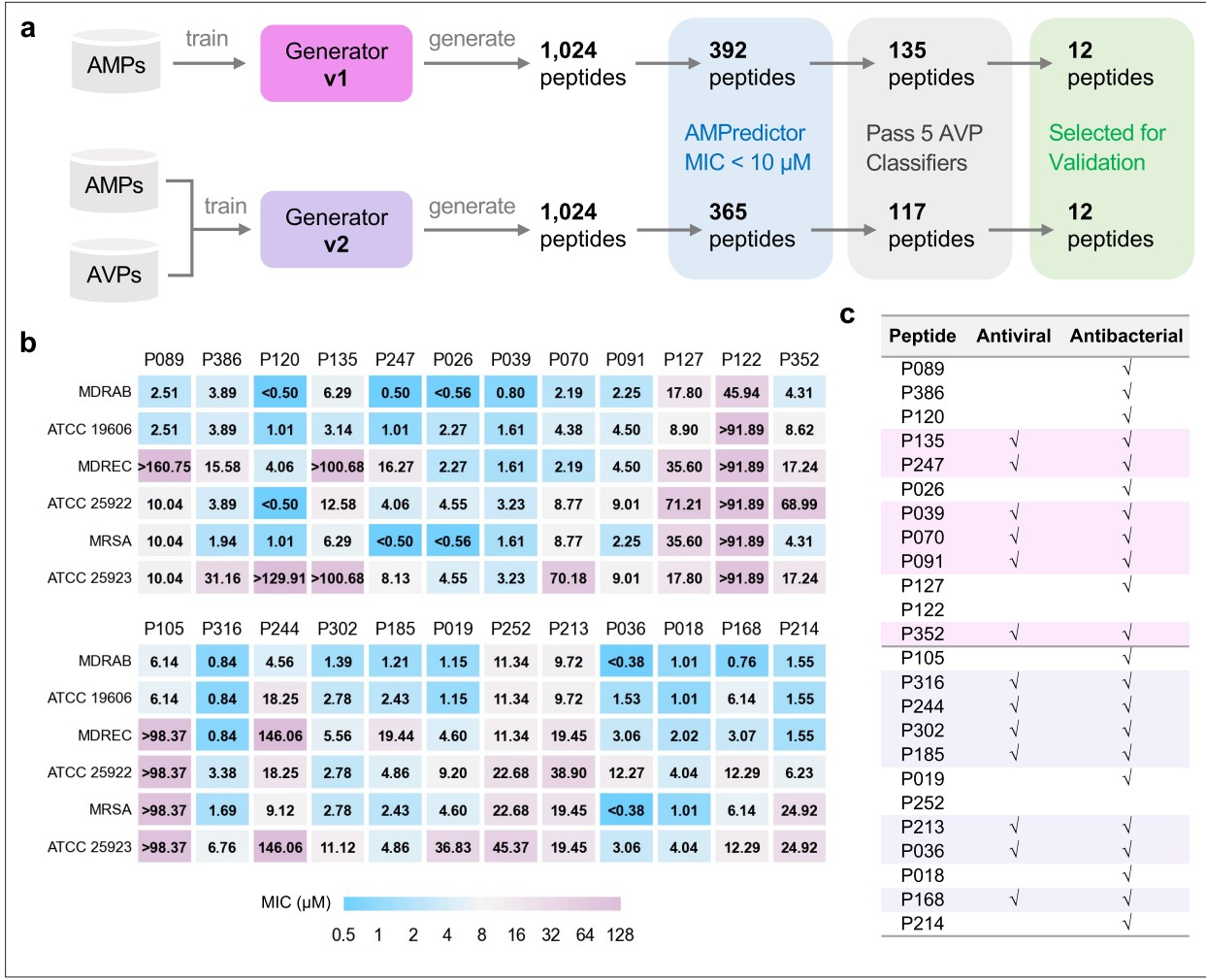

**Figure 2.** Large-scale design and validation of bifunctional antimicrobial peptides (AMPs). (**a**) Schemes of designing bifunctional AMPs from two versions of generators. 12 designed peptides from each generator were selected for experimental validation. (**b**) Minimal inhibitory concentration (MIC) test of 24 designed peptides against six bacterial strains. The first group included 12 peptides from generator v1, and the other 12 peptides were from generator v2. (**c**) Antiviral and antibacterial results for bifunctional peptides. The rows with pink or purple backgrounds are bifunctional AMPs acquired from generators v1 or v2, respectively.

The online version of this article includes the following figure supplement(s) for figure 2:

**Figure supplement 1.** Mass spectrometry analysis.

**Figure supplement 2.** Mass spectrometry of P089.

**Figure supplement 3.** Mass spectrometry of P386.

**Figure supplement 4.** Mass spectrometry of P120.

**Figure supplement 5.** Mass spectrometry of P135.

**Figure supplement 6.** Mass spectrometry of P247.

**Figure supplement 7.** Mass spectrometry of P026.

**Figure supplement 8.** Mass spectrometry of P039.

**Figure supplement 9.** Mass spectrometry of P070.

**Figure supplement 10.** Mass spectrometry of P091.

**Figure supplement 11.** Mass spectrometry of P127.

**Figure supplement 12.** Mass spectrometry of P122.

**Figure supplement 13.** Mass spectrometry of P352.

**Figure supplement 14.** Mass spectrometry of P105.

*Figure 2 continued on next page*

*Figure 2 continued*

26 were predicted to own MIC values below 10 µM and 5 µM, respectively (*Supplementary file 2*). The small-scale design procedure did not sacrifice the efficiency of discovering novel peptides and featured a comparable success rate with AMPredictor. We regarded a peptide as an antiviral candidate if four or more positive votes were recorded and 32 peptides met our criteria.

We also conducted a comprehensive evaluation of efficacy and safety in silico to select novel AMPs before experimental validations. Considering the membrane corruption mechanism of most AMPs, we predicted their hemolytic activity as a quick measure of cellular toxicity (*Timmons and Hewage, 2020*). Through the ranking of high activity and low toxicity, we picked three potent peptides P001, P002, and P076 (*Figure 1f*), all of which featured amphiphilic signatures of sequence and enriched cationic regions (e.g., helical wheel projections in *Figure 1—figure supplement 3*), for in vitro and in vivo experimental assessment.

### In vitro antibacterial and hemolytic assays

We examined the antibacterial activity of P001, P002, and P076 against one Gram-positive (*S. aureus*) and three Gram-negative (*A. baumannii*, *E. coli*, and *P. aeruginosa*) bacteria, including three ESKAPE strains. For each species, we cultured a standard strain and a MDR strain, and measured the MIC values of individual peptides using the broth microdilution method. All the peptides exhibited inspiring bactericidal properties with MICs ranging from 0.20 to 15.18 µM (corresponding to 0.625–40 µg/mL, *Figure 3a*). P001 showed broad-spectrum antibacterial capability with MICs less than 4 µM across all the eight strains of four bacteria species. P002 featured lower MIC values in inhibiting Gram-positive bacteria *S. aureus*, while P076 mainly inhibits Gram-negative bacteria *A. baumannii*, *E. coli*, and *P. aeruginosa*. Notably, the experimentally measured MICs against *E. coli* agreed well with the predictions of AMPredictor (*Figure 1f*), indicating the accuracy of our regression models.

The hemolytic analysis revealed that both P001 and P002 displayed dose-dependent toxicity to mouse erythrocytes (*Figure 3b*), with average hemolysis higher than 50% at high concentrations (20 µM, or 100 µg/mL). Nevertheless, the hemolysis of P076 was lower than 1% at concentrations up to 70 µM. We also inspected the apparent cytotoxicity ($CC_{50}$) of three peptides on three mammalian cell lines (Vero-E6, A549, Huh-7) using CCK8 assays. For the cell cultures, $CC_{50}$ values of all three peptides were well above 30 µM (*Figure 4—figure supplement 3a*). The $CC_{50}$ of P076, specifically, were in the range of 35.5 µM (for A549 cells from lung tissues) to 66.8 µM (for Huh-7 cells from liver tissues). The in vitro safety assays pointed out thresholds of concentration for administrating these peptides, which were adapted in subsequent animal experiments for estimation of safe dosage.

Our data suggest that all three peptides are potential candidates for inhibiting the clinically relevant *A. baumannii*, which is a top-ranking pathogen with critical priority for development of new antibiotics, as defined by WHO (*De Oliveira et al., 2020*). The MIC of P076 at eliminating the persistent MDRAB reached 0.21 µM, a value half of P001 and P002 (0.40 and 0.47 µM, respectively). Meanwhile, P076 was comparable with P001 but more efficient than P002 against *E. coli* and *P. aeruginosa*. Despite a slight inferiority in killing *S. aureus*, P076 owned a better biosafety. Taken together, the sequence of P076 likely leveraged the balance between efficacy and safety; we therefore focused the following in vivo evaluation and animal infection models on P076.

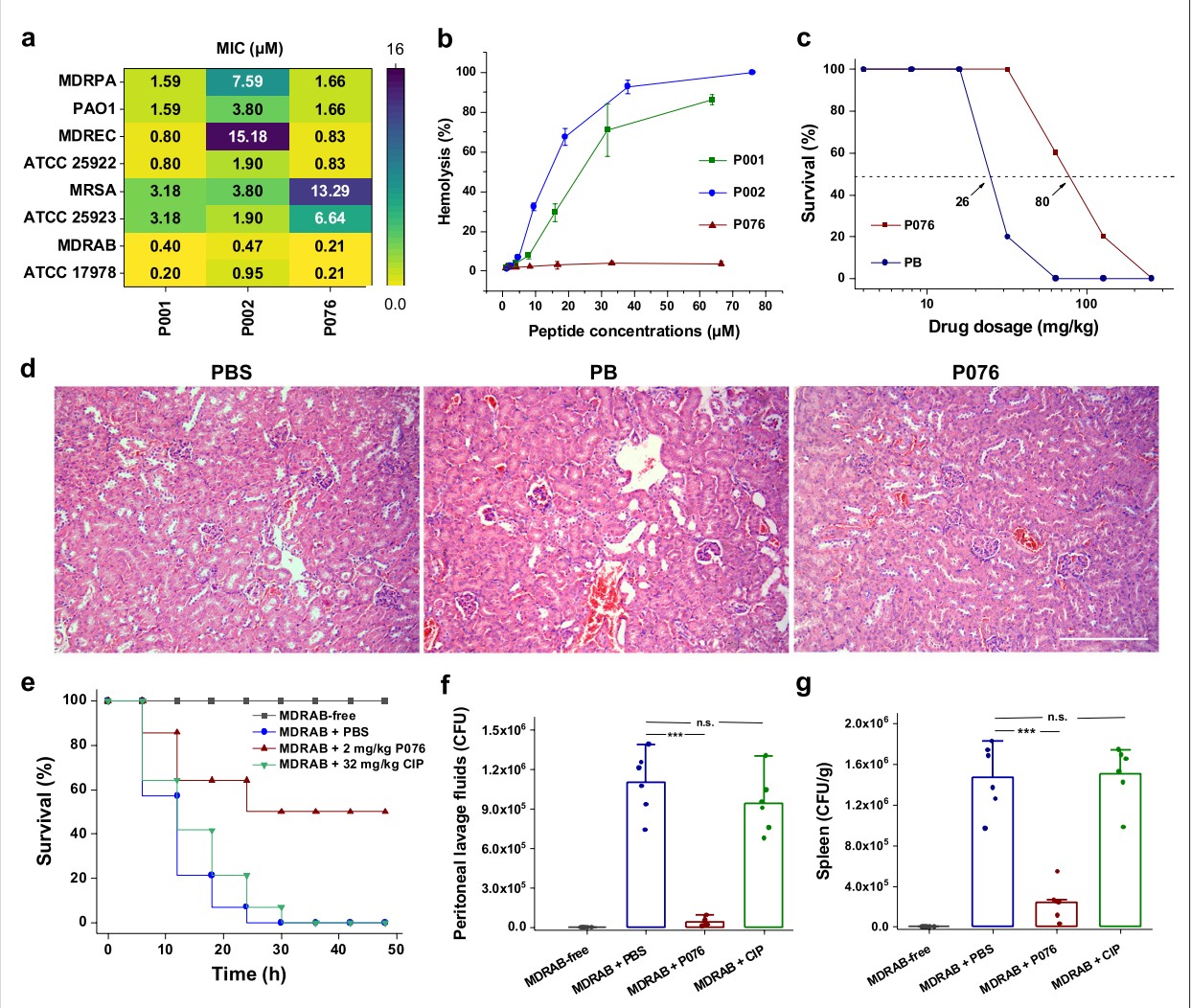

**Figure 3.** Antibacterial and toxicological evaluations of P076 peptide. (**a**) In vitro antibacterial assessment of peptides by determination of minimal inhibitory concentrations (MICs). (**b**) Hemolysis of increasing concentrations of peptides. Results are shown as the means ± SDs. (**c**) Survival curves of mice given P076 and polymyxin B (PB). Arrows indicate the drug concentration inducing a 50% death. (**d**) H&E staining of mouse kidneys. The scale bar indicates 200 μm. (**e**) Survival of infected mice treated with P076 and ciprofloxacin (CIP). (**f, g**) Bacterial colonization of multidrug-resistant *A. baumannii* (MDRAB) in mouse peritoneal lavage fluid and spleen. Results are shown as the means ± SDs. ***p<0.001. n.s., no significance.

The online version of this article includes the following figure supplement(s) for figure 3:

**Figure supplement 1.** Additional results for in vivo antibacterial evaluations of P076.

## In vivo antibacterial and toxicological evaluations

We first determined a safe dosage for administrating P076 on mouse models. A total of 140 C57 mice (randomly divided into 14 groups) were given increasing concentrations of P076 through intra-peritoneal injection, and the cyclic peptide antibiotic PB was employed as a positive control. In spite of nephrotoxicity, PB is one of the polymyxin antibiotics that is considered 'the last ditch' for the treatment of infections caused by extensively drug-resistant Gram-negative bacteria in clinic (*Chung et al., 2016*). Animal survival was monitored for 48 h after a single drug administration. The P076 concentration to induce a 50% death was calculated to be 80 mg/kg, approximately threefold higher than that of PB (26 mg/kg, *Figure 3c*). H&E staining further verified that no pathological changes were observed in the mice liver, spleen, and kidneys of mice given 32 mg/kg P076 after 12 h, whereas an apparent renal tubular injury was discovered in PB-treated mouse (*Figure 3C*, *Figure 3—figure supplement 1a*).

Encouraged by the biocompatibility of P076, we further conducted an in vivo antibacterial assay using a mouse peritoneal infection model (*Zhao et al., 2022*). A total of $3 \times 10^7$ colony-forming units (CFUs) of MDRAB was intraperitoneally injected in C57 mouse to establish infection symptoms. P076 was administered at 2 mg/kg at 0.5 and 2.5 h after bacterial invasion. As the positive control treatment, we employed a single injection of 32 mg/kg ciprofloxacin (CIP) at 0.5 h after infection. The mice given PBS (blank buffer) or CIP all deceased after 48 h, while 50% of the mice treated with P076 remained alive (*Figure 3e*). Additional investigations revealed that the bacterial colonization in the peritoneal lavage fluid of PBS-treated mouse was averagely $1.1 \times 10^6$ CFU, which was 25 times higher than that in the mice treated with P076 ($4.2 \times 10^4$ CFU, p<0.001), whereas CIP featured less of effect on the number of bacteria (*Figure 3f*). Similarly, P076 dramatically decreased the bacterial loads in the mouse spleen (about seven times lower, *Figure 3g*, *Figure 3—figure supplement 1b*), thus demonstrating the prospect of P076 to be developed into a new antibiotic.

## Broad-spectrum antiviral assays

A cohort of traditional AMPs have been rediscovered with antiviral abilities (*Liu et al., 2022*). One significant reason is that their membrane-disrupting capacity can adapt to both bacterial membranes and viral envelopes, which may be captured by AMPredictor and AVP classifier filters. Thus, we tested the antiviral activities of P001, P002, and P076, which passed our dual filters. Our experiments included five typical enveloped viruses. Chikungunya virus (CHIKV), Hantaan virus (HTNV), Dengue virus 2 (DENV-2), Herpes simplex virus 1 (HSV-1), and severe acute respiratory syndrome coronavirus 2 (SARS-CoV-2, and the BA.2. omicron variant) are all pathogens of the past or ongoing epidemics, such as the chikungunya fever, hemorrhagic fever with renal syndrome, dengue, encephalitis, and COVID-19. Hence, the peptide drugs may be repurposed as preventions against viral infections.

We incubated the peptides with different viruses before they invaded the cultured cells and self-amplified. We analyzed the potential antiviral effects of P001, P002, P076 at the mRNA and protein levels, respectively. At the mRNA level, the quantitative real-time PCR (qRT-PCR) suggested dose-dependent inhibition of viral-specific gene expressions (E1 gene for CHIKV, S gene for HTNV, VP16 gene for HSV-1, NS5a gene for DENV-2, ORF for SARS-CoV-2, *Figure 4a*, *Figure 4—figure supplement 1b*). At the protein level, we applied fluorescence-conjugated antibodies to visualize trace amounts of virus populations. Corroborating with the qRT-PCR results, P002 nearly abolished all viral infections at 12.5 μM (*Figure 4b*, *Figure 4—figure supplements 1a and 2*). To quantify the immunofluorescence results, we plotted the log-scale fluorescent focus forming unit (FFU) against varying peptide concentrations (*Figure 4—figure supplement 3b*). Among the three peptides, P002 exhibited the highest potential with $EC_{50}$ of 0.37 μM (against CHIKV) to 2.08 μM (against HTNV) (*Figure 4b*, *Supplementary file 1d*). The $EC_{50}$ of P076 and P001 were in the range of 1.62–2.67 μM (except for inhibiting DENV-2).

Additionally, we tested the antiviral effects of P135 and P244 from our second round of generation. Both the immunofluorescence and qRT-PCR results showed that these AMPs were active broad-spectrum AVPs against four viruses (*Figure 4—figure supplement 4* and *Figure 4—figure supplement 5*). In particular, these two AMPs showed a significant inhibition on DENV-2, for which P135 obtained an $EC_{50}$ of 0.09 μM (*Supplementary file 1d*). Hence, we demonstrated the effectiveness of GAN and AMPredictor combined pipeline on discovering new AVPs.

Combining the $EC_{50}$ and $CC_{50}$ calculations, we calculated the selectivity index (SI = $CC_{50}/EC_{50}$) as the experimental guides of safety versus efficacy (*Supplementary file 1e*). The SI of P002 against four viruses are 149.21, 31.75, 51.05, and 59.01, three of which outperformed the SI of P001 and P076, especially for the case of HTNV. Hence, we further observed the performance of P002 with transmission electron microscopy. For the control set, the viruses were allowed to propagate in cells for days and identified as assembled particles in cytosol (*Figure 5a and b*). When the viruses were mixed with 10 μM P002 before addition to the cells, the viral particles became absent while the cell morphology kept intact, indicating the broad-spectrum inhibition effects of P002 (*Figure 5c and d*).

Because we observed a gradient of antimicrobial and antiviral activities from P001, P002, and P076 with their individual preferences against bacteria and viruses, we aimed to probe the interactions between peptides and different phospholipid assemblies, thus deriving feasible means judging their underlying preferences. We performed all-atom molecular dynamics simulations with molecular models of bacterial membranes and viral envelopes, placed each peptide adjacent to the phospholipid

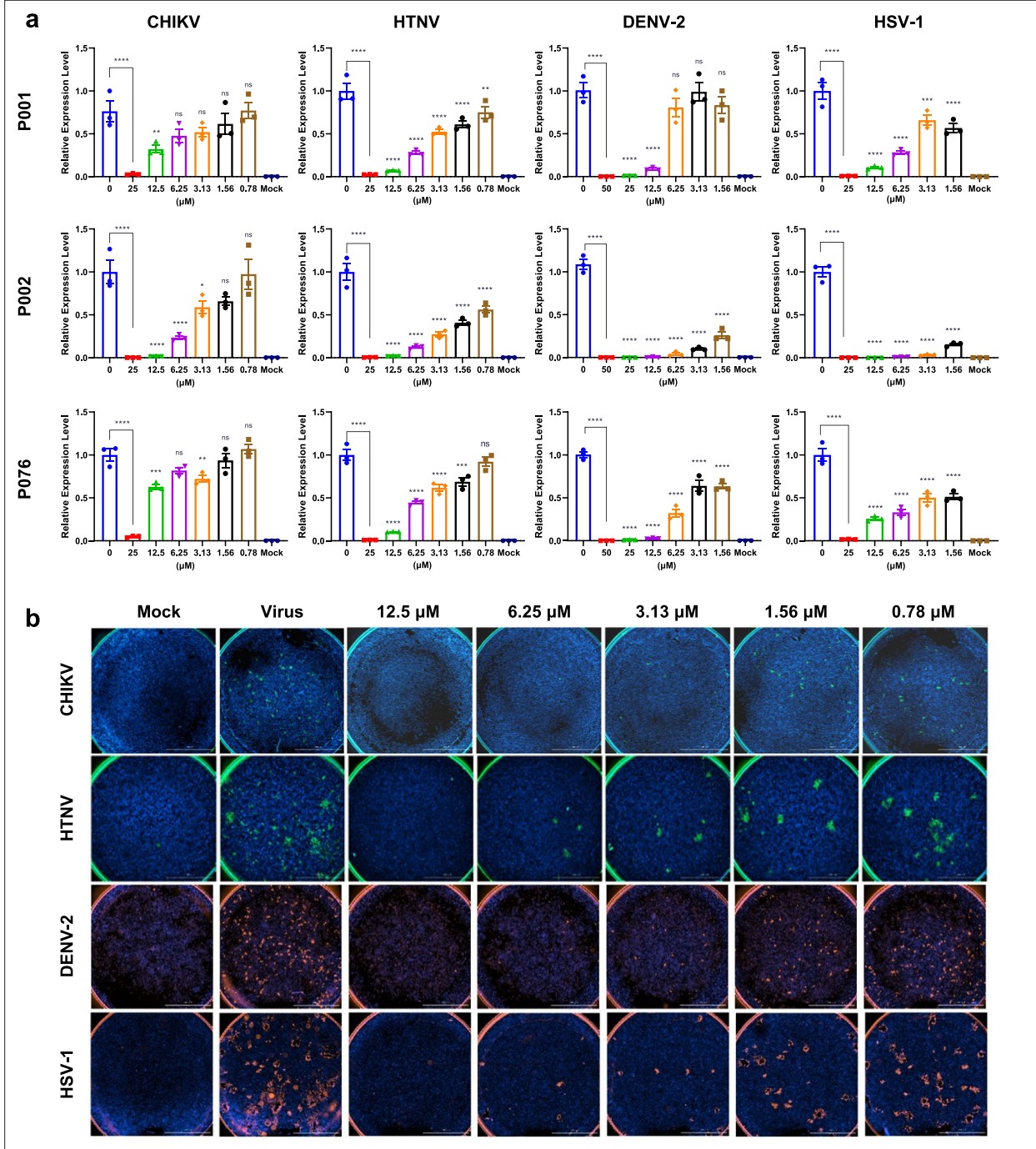

**Figure 4.** Antiviral assays of antimicrobial peptides (AMPs) against Chikungunya virus (CHIKV), Hantaan virus (HTNV), Dengue virus 2 (DENV-2), and Herpes simplex virus 1 (HSV-1). (**a**) Quantitative real-time PCR of virus RNA with gradient concentrations of P001, P002, and P076. (**b**) Immunofluorescence results of four viruses with gradient concentrations of P002. Scale bar = 2000 μm. All experiments are performed in triplicates. Column bars are means ± SEMs. *p<0.05, **p<0.01, ***p<0.001, ****p<0.0001. ns, no significance.

The online version of this article includes the following figure supplement(s) for figure 4:

**Figure supplement 1.** Antiviral assays of peptides against SARS-CoV-2 wild-type and BA.2 strains.

**Figure supplement 2.** Immunofluorescence assays of P001 and P076 inhibiting Chikungunya virus (CHIKV), Hantaan virus (HTNV), Dengue virus 2 (DENV-2), and Herpes simplex virus 1 (HSV-1).

**Figure supplement 3.** Cytotoxicity and quantitative immunofluorescence results.

*Figure 4 continued on next page*

*Figure 4 continued*

**Figure supplement 4.** Immunofluorescence assays of P135 and P244 inhibiting Chikungunya virus (CHIKV), Hantaan virus (HTNV), Dengue virus 2 (DENV-2), and Herpes simplex virus 1 (HSV-1).

**Figure supplement 5.** Quantitative real-time PCR of virus RNA with gradient concentrations of P135 and P244 inhibiting Chikungunya virus (CHIKV), Hantaan virus (HTNV), Dengue virus 2 (DENV-2), and Herpes simplex virus 1 (HSV-1).

bilayers, and simulated them attaching to the membrane (*Figure 5—figure supplement 1a and b*). For the model of Gram-negative bacteria systems, the MM/PBSA calculations and $\Delta G_{bind}$ distinguished P002 from P076 and P001 (*Supplementary file 1j*). Evaluated by $\Delta G_{bind}$ (–465.79 ± 11.39 kJ/mol), P076 would associate with the membrane of Gram-negative bacteria tighter than others. For the model of viral envelopes, all binding energies were weaker than that to bacteria. The ranking of $\Delta G_{non\text{-}polar}$ corroborated with the antiviral potential of P002. Combining this energy term with a geometrical parameter (average distance between the center of mass [COM] of peptide and phosphorus atoms

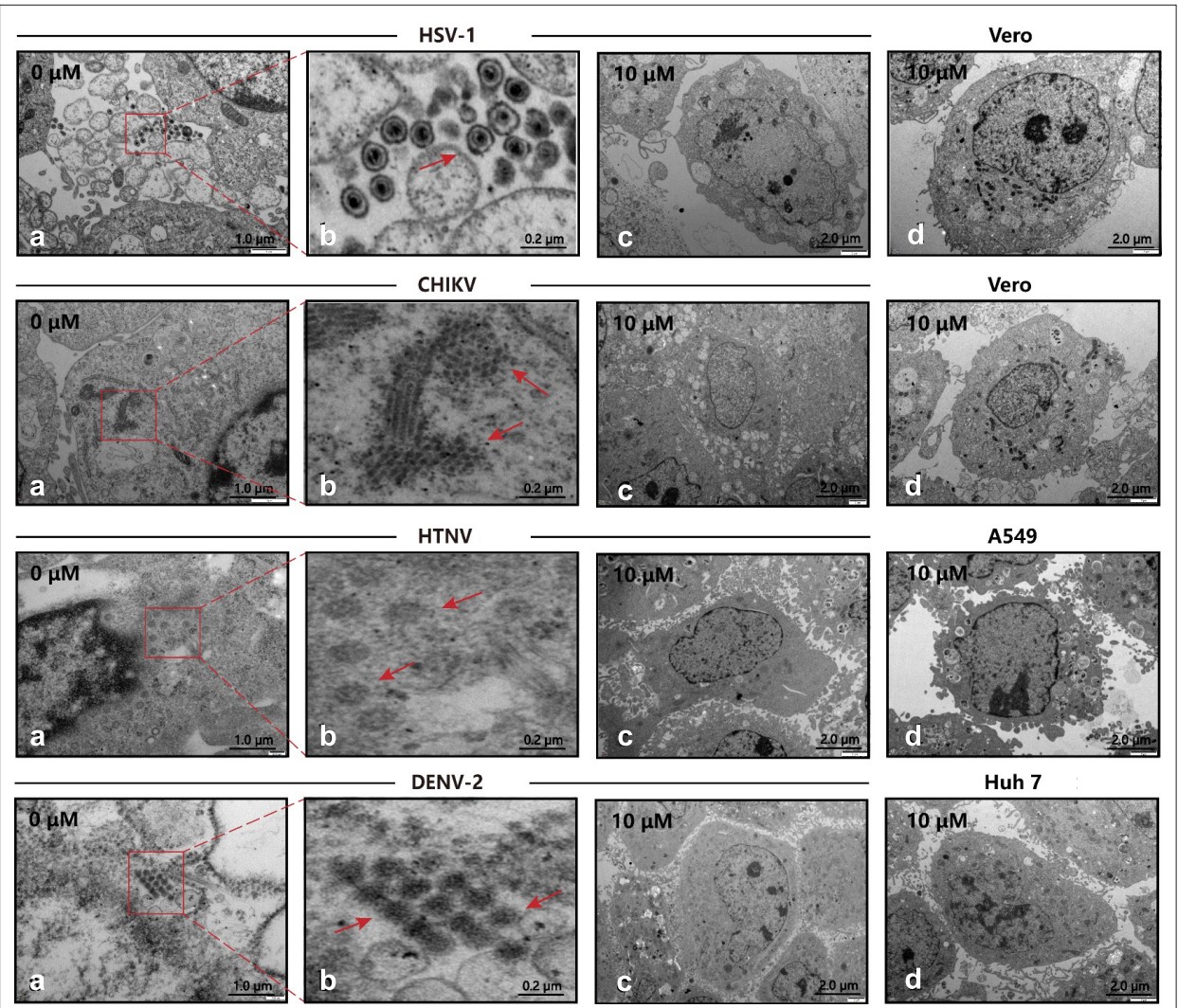

**Figure 5.** Transmission electron microscopy (TEM) of viral infection with P002. (**a, b**) Cells infected with virus, where the red arrows mark the viral particles. (**c**) Cells are treated with 10 µM P002 and infected with virus (multiplicity of infection [MOI] = 1), where no viral proliferation is observed. (**d**) Control of intact cells with 10 µM P002. All experiments are performed in triplicates.

The online version of this article includes the following figure supplement(s) for figure 5:

**Figure supplement 1.** Molecular dynamics simulations for peptides and different lipid bilayers.

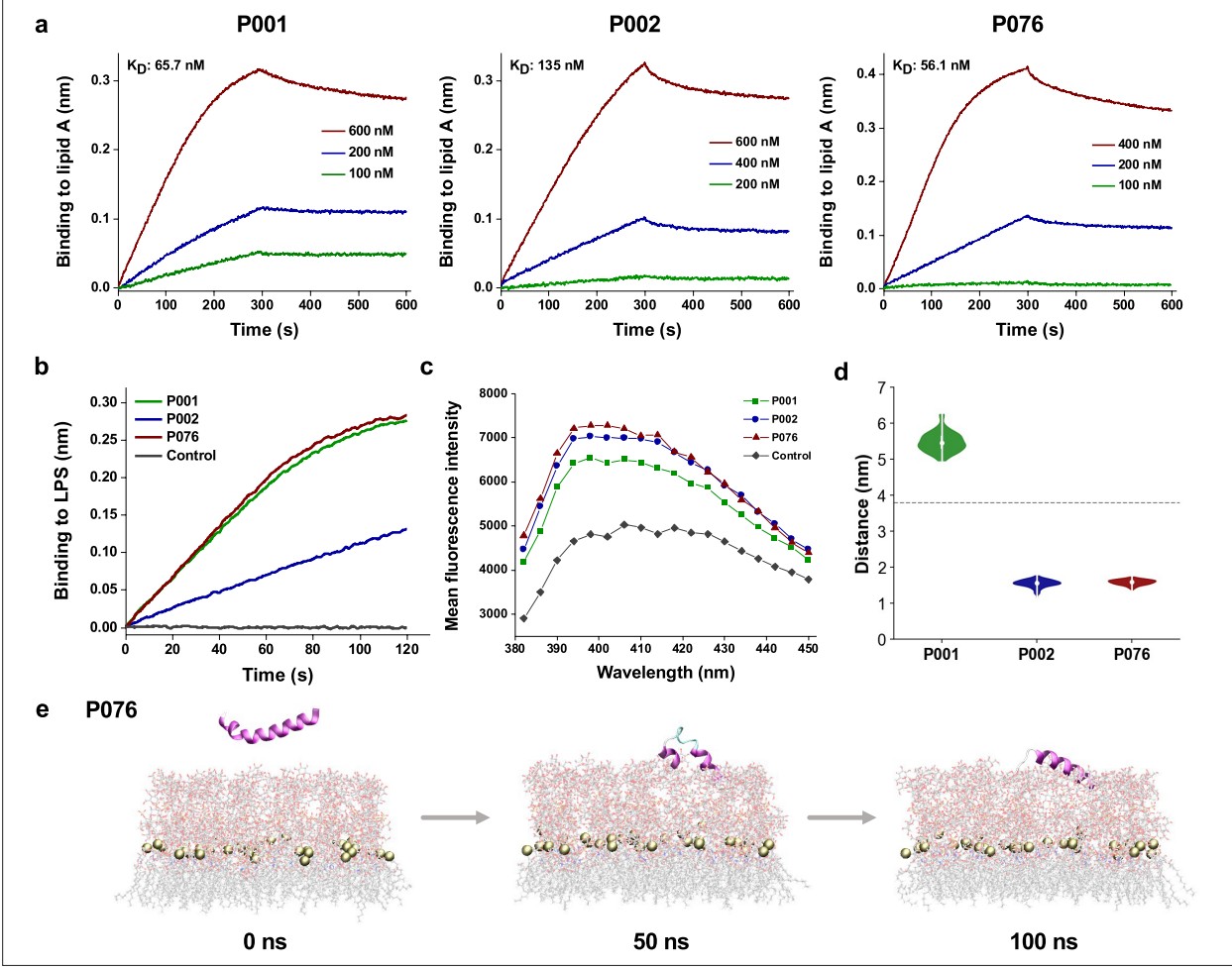

**Figure 6.** Bacterial membrane-acting assessment of antimicrobial peptides (AMPs). (**a**) Binding kinetics of three AMPs to lipid A measured by biolayer interferometry. (**b**) Binding thicknesses of 500 nM AMPs to immobilized lipopolysaccharide (LPS) within 120 s. (**c**) Fluorescence intensity of 1-N-phenylnaphthylamine (NPN) after excitation at 350 nm. The maximum emission appeared at approximately 396 nm, where P001, P002, and P076 were detected higher intensity than control. (**d**) The distance between the center of mass (COM) of peptides and phosphorus atoms of lipid A (around the midplane of G-bacterial outer membranes) during the last 50 ns of the molecular dynamics (MD) simulation trajectories. The gray dashed line displays the initial distance at 0 ns. (**e**) Snapshots of P076 with G- bacterial outer membranes at 0 ns, 50 ns, and 100 ns. Only the top leaflet (LPS) is shown. Tan balls represent the phosphorus atoms of lipid A molecules.

in the top leaflet on z-axis) may predict the behavior of P002 better than $\Delta G_{nonpolar}$ alone (*Figure 5—figure supplement 1c*).

## Membrane-penetration mechanisms

We hypothesized that a common target among bacteria and viruses was the membrane components. To shed light on the possible membrane-penetration abilities of designed peptides, we applied quantitative measurements, first by focusing on signature components of Gram-negative bacterial outer membranes. We measured the binding of three peptides to lipopolysaccharide (LPS) via biolayer interferometry (BLI). More P076 and P001 peptides were recruited to immobilized LPS than that of P002 at 500 nM (*Figure 6b*). Further experiments demonstrated that P076 bound to lipid A, the innermost part of LPS, at an equilibrium dissociation constant ($K_D$) of 56.1 nM. This binding affinity of P076 to lipid A was slightly higher than that of P001 (65.7 nM) and was approximately 2.4-fold less than that of P002 (135 nM, *Figure 6a*). Then we directly investigated the membrane permeability by 1-N-phenylnaphthylamine (NPN) uptake assay. NPN is a fluorescent dye for detecting the impaired outer membrane of Gram-negative bacteria. P076 elicited a stronger membrane penetration relative

to P001 and P002 (wavelength scanning in *Figure 6c*). Together, our results suggested that P076 could efficiently bind to and damage bacterial membranes.

To gain a more comprehensive understanding of AMP-membrane interactions with fine content of amphiphilic assemblies (both outer membranes and inner membranes), we also observed the membrane attachment and partial insertion of AMPs via all-atom molecular dynamics (MD) simulations. The top leaflet of Gram-negative bacterial outer membranes was built as simplified *E. coli* LPS (lipid A and R1 core). With an initial equilibrated position above the membrane surface, the peptide started to approach the oligosaccharides of LPS and then bind to them. During the simulation processes, P002 and P076 gradually inserted into the outer leaflet and represented poses binding to LPS at 100 ns (*Figure 6e*). We also measured the distance between the center of mass of peptide and phosphorus atoms of lipid A on z-axis in the latter 50 ns of trajectories (*Figure 6d*). Compared with the initial value about 4 nm (showing as dashed line), this metric of P002 and P076 declined to less than 2 nm. Nevertheless, P001 performed weaker interactions with an increasing distance, which was consistent with the membrane-disrupting experiment in *Figure 6c*. Moreover, three AMPs tended to attach to the membrane in the MD simulations with lipid bilayer system mimicking the inner membranes of Gram-negative bacteria (*Figure 5—figure supplement 1b*). Collectively, experimental characterizations along with computational simulations indicate the membrane-acting mechanisms of our AMPs when playing antimicrobial roles. Further explanation of membrane-disrupting details or other potential biomolecular targets remains to be explored in the future.

## Discussion

The traditional 'lock and key' model for therapeutic molecule design requires knowledge of target structures (*Kitchen et al., 2004*). When the targets are biological membranes, the means of docking and ranking by binding affinities ($K_D$) can easily become futile. The rapid development of deep neural networks, on the other hand, has facilitated self-supervised learning of meaningful molecule representations in hidden dimensions (*Rives et al., 2021*). Combined with vast data of known AMPs, numerous deep learning models have enabled mining of new antibiotics (*Maasch et al., 2023*; *Torres et al., 2022*). In the current study, we deployed a new combination of descriptors to extract representations of biological sequences at residual and protein levels. First, we transformed input amino acid sequences into Morgan fingerprints to encode the chemical features of each residue. Second, the scaffolds of AMPs are closely relative to their antimicrobial function, but rarely considered at the structural level in recently reported models (except for sAMPpred-GAT *Yan et al., 2023*). Instead of using limited amounts of crystal structures, we introduced predicted contact maps of peptides as the edge encoding of peptide graphs. Finally, we added the embedding from ESM as the residue node features to present the associations of a single peptide sequence in the large protein repertoire. ESM was trained on millions of natural protein sequences via mask attention mechanism and assisted in few-shot learning via transferring the related knowledge of protein sequences (*Rives et al., 2021*). The resulting AMPredictor accurately predicted the MICs of three de novo design AMPs in our study. As an independent test on 28 new AMPs discovered by different models, AMPredictor also scored well (*Supplementary file 1f*).

Despite the ability of deep learning model to guide targeted generation and screen of functional peptides, it lacks the merit of molecular docking program to explicitly consider subtle environmental features. For a particular sequence, we found it was difficult to tell its preference toward bacterial membranes or viral envelopes due to slight differences in their lipid compositions. Consequently, we observed broad antimicrobial activity for P001, broad antiviral activity for P002, while P076 demonstrated superior capacity on *A. baumannii*. As a control, we made a tiny change of P076 C-terminal with amide modification but observed loss of functions (*Supplementary file 1g*). The change of a few atoms on an intact sequence can lead to dramatic functional consequences, which may be challenging for neural networks to capture.

Recent studies indicated unexpected side effects of traditional antibiotics. Instead of restraining the growth of carbapenem-resistant *Enterobacteriaceae* in the intestine environment, broad-spectrum antibiotics reduced gut microbial populations and depleted beneficial metabolites, thus facilitated colonization of *Enterobacteriaceae* species (*Yip et al., 2023*). With more limitations of conventional small molecule antibiotics discovered, AMPs are attractive alternatives to conventional antibiotic treatment. Their wide bioactivity spectra can expand to virus, thus minimizing the cost of screen

multiple drugs. Our results suggested efficient in silico screen tools plus rigorous experimental validations could uncover highly potent peptide drugs without comprising on safety, and large-scale animal experiment could facilitate translation toward clinical research. In summary, we believe that a mature workflow involving sequence generation and evaluation can promote the discovery of ideal AMPs to eventually alleviate the hazard of drug-resistant diseases.

# Materials and methods

## Data curation

Our AMP dataset is collected from PepVAE (*Dean et al., 2021*) , containing 3280 AMP sequences with their MICs in logarithmic form. To integrate various AMP sequence and bioactivity data, Witten et al. developed the Giant Repository of AMP Activity (*Witten and Witten, 2019*). Since the MIC values change when targeting different microbial strains, only AMPs inhibiting *E. coli* are selected. Also, to simplify the input and reduce accidental effect, peptides without modifications are filtered so that no cysteine appears (amino acid composition in *Figure 1—figure supplement 1a*). The sequence length of AMPs is less than 40 amino acids, and the majority of lengths (~60%) are between 10 and 20 (*Figure 1—figure supplement 1a*). Labels, that is, logMIC values, are distributed from –1 to 4. During the training process, the dataset is divided into the training, validation, and testing sets in a ratio of 8:1:1 randomly. The entire AMP dataset (3280 sequences) is used for training the generator v1. The AVP dataset for training the generator v2 is collected from AVPdb (accessed in August 2023). AVPs with lengths of more than 60 amino acids are excluded. After removing overlaps with the AMP dataset, 1788 AVPs are kept and added to the training set of generator v2.

## Generator

The GAN is composed of a generator network and a discriminator. Through the co-training strategy, the generator evolves to produce ideal sequences from random noises *z*, fitting the distribution of real samples and faking the discriminator.

$$min_G max_D V(G, D) = E_{x \sim p_{real(x)}} \left[ logD(x) \right] + E_{z \sim p_{fake(z)}} \left[ log\left(1 - D\left(G(z)\right)\right) \right] \tag{1}$$

For the combined training of GAN generator and discriminator, the AMP sequences were encoded with AAFs. AAF encoding represents five physiochemical values, including the polarity, secondary structures, molecular size, codon diversity, and charges (*Atchley et al., 2005*). The GAN architecture was utilized to generate novel AMP sequences, where three layers of 2D convolution were inside both the generator and discriminator (*Tzu-Tang et al., 2021*). Loss function modified with the Wasserstein GAN with gradient penalty (WGAN-GP) strategy was taken to avoid mode collapse, and the optimizer was kept as RMSProp. The learning rate was 1e-4. Cross training ratio between generator and discriminator was set as 1:5. Deep learning-based models were trained on an NVIDIA GeForce RTX 2080 Ti GPU. After training for 1000 epochs, string sequences are calculated via cosine similarity. A batch of sequences were generated with the limited length of 30 amino acids. Redundant alanine (A) at the end of every sequence were removed, thereby leaving sequences with variable length.

## Predictor

AMPredictor was built based on PyTorch and PyTorch Geometric library. All input sequences were treated with padded zeros to reach unified lengths of 60. For chemical fingerprint, RDKit (https://www.rdkit.org/) was used to transform each amino acid into 2048-bit Morgan fingerprint in peptide sequences. For each peptide, the fingerprint vectors were concatenated and contrasted into a 60-dimensional vector by average (*Schissel et al., 2021*). Pretrained language model esm1b_t33_650M_UR50S was used to provide the embeddings and the contact maps.

During the training process, the optimizer was Adam, and loss function was the mean squared loss. Batch size and learning rate were decided upon grid-search, whose values were 128 and 0.001, respectively (*Figure 1—figure supplement 1c*). Evaluation metrics mean square error (MSE), RMSE, PCC, and concordance index (CI) are calculated as follows:

$$MSE = \frac{1}{n} \sum_{1}^{n} (b_i - a_i)^2 \tag{2}$$

$$RMSE = \sqrt{\frac{1}{n} \sum_{1}^{n} (b_i - a_i)^2} \tag{3}$$

$$PCC = \frac{\sum_{i=1}^{n} (a_i - \bar{a})(b_i - \bar{b})}{\sqrt{\sum_{i=1}^{n} (a_i - \bar{a})^2 \sum_{i=1}^{n} (b_i - \bar{b})^2}} \tag{4}$$

$$CI = \frac{1}{Z} \sum_{a_i > a_j} h(b_i - b_j) \tag{5}$$

where $a_i$ is the real activity label of peptide $i$, and $b_i$ is the predicted activity value, $Z$ is the scale constant, and $h(b)$ is the step function.

For comparisons in *Figure 1—figure supplement 2*, five baseline models were implemented using DeepPurpose package (*Huang et al., 2021*). The amino acid composition encoding was used as the input of MLP, while one hot encoding was for CNN, CNN-GRU, and CNN-LSTM. Fragment partition fingerprint of peptides was fed into the Transformer encoder.

In ablation study, three encoding approaches (chemical fingerprint, contact matrix, and ESM) and their different combinations were tested individually. It was noted that the peptide graph exists only when both the node embedding and contact matrix are present together. Therefore, a CNN with the same number of layers and output dimension was adopted instead of the GCN in ablation test.

## Peptide synthesis

All the peptides were synthesized by DGpeptide Co., Ltd. with 95% purity via solid-phase peptide synthesis procedures. Mass spectrometry was used to verify the molecular weights (*Figure 2—figure supplements 1–25*).

## MIC detection

The minimum inhibitory concentrations (MICs) of peptides against bacteria including *A. baumannii* (ATCC 17978), multidrug-resistant *A. baumannii* (MDRAB) (*Wang et al., 2018*), *S. aureus* (ATCC 25923), methicillin-resistant *S. aureus* (MRSA, ATCC 43300), *E. coli* (ATCC 25922), multidrug-resistant *E. coli* (MDREC) (*Zhao et al., 2019*), *P. aeruginosa* (PAO1), and multidrug-resistant *P. aeruginosa* (MDPRA) (*Gao et al., 2020*) were determined using the broth microdilution method recommended by the Clinical and Laboratory Standards Institute with modifications reported by the Hancock Laboratory (*Wu and Hancock, 1999*). Peptides were prepared in 0.01% acetic acid containing 0.2% BSA at concentrations of 6.25, 12.5, 25, 50, 100, 200, 400, 800, 1600, and 3200 µg/mL. Bacteria grown to the mid-logarithmic phase in Meller–Hinton broth (MHB) were diluted to $2 \times 10^5$ CFU/mL. Aliquot of 100 µL of the bacterial suspension was then co-incubated with 11 µL of the peptides at 37°C in a sterile polypropylene microtiter plate (3365, Corning, Shanghai, China). An M2e microplate reader (Molecular Devices, Silicon Valley, CA) was applied to detect the bacterial absorbance at 600 nm after 18 h. The recorded MIC was the lowest concentration of peptides, causing a 70% reduction in bacterial turbidity.

## Hemolytic assay

The hemolysis of peptides was evaluated as previously described in *Oddo and Hansen, 2017*. Briefly, the fresh mouse blood was centrifuged at $1000 \times g$ for 10 min and washed with cold PBS three times. The erythrocytes were collected and diluted in saline solution to prepare a 2% (v/v) suspension. A total of 75 µL of the erythrocyte suspension was co-incubated with 75 µL of the peptides (6.25, 12.5, 25, 50, 100, 200, and 400 µg/mL) at 37°C for 1 h. After centrifuging at $1000 \times g$ for 10 min, 60 µL supernatant was obtained, and the absorbance was determined at 414 nm. This experiment was conducted in triplicate and repeated twice.

## In vivo antibacterial evaluation

The therapeutic efficacy of P076 peptide against drug-resistant bacteria was analyzed using a mouse peritoneal infection model (*Zhao et al., 2022*). The 8-week-old female C57 mice (n=10) was

intraperitoneally administered with $3 \times 10^5$ CFU/mL of MDRAB. P076 (2 mg/kg) was given by intraperitoneal injection at 0.5 and 2.5 h after bacterial infection. CIP (C129896, Aladdin, 32 mg/kg) was employed as the control. Mouse survival was monitored for 48 h. For detection of the bacterial colonization, another 24 mice were randomly divided into four groups (n=6 for each group). The mice were euthanized 12 h after MDRAB infection, and the peritoneal cavity was rinsed with 2 mL of PBS. Additionally, mouse liver and spleen were obtained by surgery and homogenized in 1 mL of PBS. The lavage fluid and tissue suspension were both diluted for bacterial colony counting using an MHB agar plate as described in the section 'MIC detection'.

## In vivo toxicological evaluation

The 8-week-old female mice (18–22 g) were randomly divided into 14 groups (n = 10 for each group). P076 peptide and PB (P105490, Aladdin, Shanghai, China) were given to the mice by intraperitoneal injection at 4, 8, 16, 32, 64, and 128 mg/kg. Mouse survival was monitored for 48 h. For pathologic analysis, mouse liver, spleen, and kidneys were obtained by surgery for H&E staining after treatment with 64 mg/kg P076 and PB for 12 h. Mice were cared for and treated in accordance with the NIH guidelines for the care and use of laboratory animals (NIH publication no. 85e23 Rev. 1985). Animal experiments were conducted with approval from the Animal Experimental Ethics Committee of the Army Medical University (AMUWEC20223290).

## Cell cultures

Vero-E6 (African green monkey kidney cell line), Huh-7 (hepatocellular carcinoma cell line), and A549 (human lung carcinoma cell line) were purchased from the National Collection of Authenticated Cell Cultures (serials: SCSP-520, SCSP-503, and SCSP-526). The STR typing results of the cell DNA showed clear amplification profiles and reliable typing results, and the mycoplasma detection result was negative. Vero-E6, A549, Huh-7, and HeLa-hACE2 were cultured in DMEM (Thermo Fisher Scientific, Waltham, USA) containing 10% fetal bovine serum in a 5% $CO_2$ incubator at 37°C. HSV-1 and CHIKV were propagated in Vero-E6 cells. DENV-2 and HTNV were propagated in Huh-7 and A549 cells, respectively. SARS-CoV-2 were propagated in HeLa-hACE2 cells.

## Cytotoxicity measurements

The Vero-E6, Huh-7, and A549 cells ($2 \times 10^4$ cells/well) were inoculated in the 96-well culture plate and grown overnight to 90% confluence at 37°C. The cells were treated with different concentrations of peptides (P001, P002, and P076) for 1 h, and the same concentration of PBS was used as a control. After 48 h of cell culture, 10% CCK8 solution was added to each well and incubated at 37°C for 1 h. The absorbance was measured at 450 nm using a microplate reader. The $CC_{50}$ represents the peptide concentration at the 50% of viability of uninfected cells.

## Immunofluorescence assay

The cells were cultured in a 96-well plate ($2 \times 10^4$ cells/well). Different viruses were mixed with P001, P002, or P076 peptides for 10 min before infecting the cells for 1 h. The infected cells were then cultured with DMEM containing 10% serum for 24–48 h. After discarding the supernatant and washing it three times with PBS, it was permeated with PBS containing 0.2% Triton X-100 (containing 0.1% BSA) at room temperature for 30 min. Then, the cells were mixed with an antiviral antibody at 4°C overnight. After washing with PBS three times, the cells were incubated with fluorescence-conjugated secondary antibodies at room temperature for 1 h. Cell nuclei were stained using 4',6'- diamidino-2-phenylindole (DAPI) in the dark for 5 min, and images were obtained by fluorescence microscope.

## Real-time PCR

The cells were cultured in a 6-well plate ($2 \times 10^5$ cells/well) following the same cell culture procedures as in the immunofluorescence assay. Cellular RNAs were extracted and purified using a viral RNA extraction kit (QIAGEN, Dusseldorf, Germany), and then cDNA was synthesized and monitored with SuperScript-III kit (Takara, Dalian, China). qRT–PCR analysis was performed using SYBR Green (Bio-Rad, CA) according to the manufacturer's protocol. Viral RNA expression was calculated using the $2^{-\Delta\Delta C.T.}$(cycle threshold) method normalized to GAPDH expression. The qRT–PCR primers sequences

for HTNV, CHIKV, HSV-1, and DENV-2 are listed in *Supplementary file 1h*, while the detection kits (Daan gene, Guangzhou, China) were used for SARS-CoV-2 wild-type and BA.2 subtype.

## Transmission electron microscopy

The cells were cultured in a 6-well plate ($2 \times 10^5$ cells/well) following the same cell culture procedures as in the immunofluorescence assay. The virus-infected cells were collected by centrifugation to prepare cell particles and fixed in cacodylate sodium buffer containing 1% glutaraldehyde (0.2 M, pH 7.2). The fixed sample was dehydrated by acetone solution and embedded in epoxy resin. The sample was polymerized at 60°C for 3 days. The resin block was used to prepare ultrathin slices (50–70 nm thick). These sections were supported by copper mesh, negatively stained with uranyl acetate and lead citrate (electron microscopy), and observed with a JEM100SX transmission electron microscope (JEOL, Tokyo, Japan).

## Biolayer interferometry

The Octet Red96 BLI platform (Sartorius BioAnalytical Instruments, Bohemia, USA) was utilized to measure the binding kinetics of three peptides to lipid A (L5399, Sigma, Shanghai, China). Peptides were diluted with 75 mM NaCl solution at concentrations of 100, 200, 400, and 600 nM. The amine reactive second-generation (AR2G) biosensors were processed with 1-ehtyl-3-(3-dimethylaminopropy) carbodiimide hydrochloride (E1769, Sigma) and sulfo-N-hydroxysulfosuccinimide (56485, Sigma). Subsequently, lipid A was loaded onto the activated AR2G biosensors at 30°C for 5 min, and both the processes of association and disassociation were monitored for 5 min to measure the kinetic constants. The equilibrium dissociation constant ($K_D$) was defined as the division of dissociation rate constant ($K_{off}$) and the association rate constant ($K_{on}$). In case of LPS, AR2G biosensors immobilized with 15 µg/mL LPS (L2880, Sigma) were incubated with 500 nM peptides for 2 min in 75 mM NaCl solution. The binding thickness was recorded and analyzed using ForteBio Data Analysis 7.0 software.

## NPN uptake assay

MDRAB in mid-logarithmic phase was diluted to $1 \times 10^8$ CFU/mL in 10 mM sodium phosphate buffer (pH 7.4). Then, 10 µL of each peptide was mixed with 190 µL of MDRAB at 37°C for 1 h. Afterward, NPN (104043, Sigma) was added with a final concentration of 10 µM. The fluorescence intensity was assessed with the wavelength ranging from 380 to 450 nm at 2 nm intervals after excitation at 350 nm using a Tecan Infinite M1000 Pro microplate reader (Mannedorf, Zurich, Switzerland). Each experiment was conducted in duplicate and repeated three times.

## Molecular dynamics simulations

The initial structures of peptides were built via AlphaFold2 single-sequence version (*Mirdita et al., 2022*). Lipid bilayer models were constructed using CHARMM-GUI server, along with placing parallel peptides approximately 2 nm above the membrane surface (*Wu et al., 2014*). Nine systems were built for three peptides versus three kinds of lipid bilayers (viral envelopes, Gram-negative bacterial inner membranes and outer membranes) (*Liu et al., 2022*; *Allsopp et al., 2022*). Ions of 150 mM NaCl were added to neutralize the system, and the calcium ions were used for LPS in outer leaflet of bacterial outer membranes. Detailed system settings are recorded in *Supplementary file 1i*. The standard CHARMM-GUI equilibration protocols including NVT and NPT ensembles were adopted, keeping the temperature at 310 K. All-atom MD simulations were performed using GROMACS 2021.2 with the CHARMM36 force field and the TIP3P water model for 100 ns, with a time step of 2 fs (*Best et al., 2012*; *Klauda et al., 2010*). Three independent runs were performed for each system. The visual molecular dynamics software version 1.9.4 was used for visualization.

The distance between the COM of peptides and phosphorus atoms in top leaflets was calculated via *gmx distance*. Binding free energy was calculated using the molecular mechanics Poisson-Boltzmann surface area (MM/PBSA) method on average of extracted 20 frames from the last 20 ns. The binding free energy of a peptide-membrane complex is calculated as follows:

$$\Delta G_{bind} = G_{complex} - \left( G_{peptide} + G_{membrane} \right) \tag{6}$$

$$G_{bind} = G_{polar} + G_{nonpolar} = \left( E_{elec} + G_{psolv} \right) + \left( E_{vdw} + G_{sasa} \right) \tag{7}$$

where $E_{elec}$ is the Coulomb potential, $E_{vdw}$ is the van der Waals energy, and $G_{psolv}$ and $G_{sasa}$ are polar and nonpolar solvation free energy. Following a previous study, the dielectric constant was set as 7 for phospholipids and the entropy term was neglected (*Lee et al., 2016*).

## Acknowledgements

We thank Prof. Chen Song from the Peking University for helpful discussion in MD. We also thank Qiaozhen Meng, Yuqian Pu, and Nan Song from Tianjin University. This work was supported by the National Key Research and Development Program of China (grant no. 2020YFA0908500), the National Natural Science Foundation of China (grant nos. 22007071, 31971127, 82272330, 32100122, 62322215), Natural Science Basic Research Program Key Projects of Shaanxi Province (2024JC-ZDXM-42), Shanghai Sailing Program (21YF1457800), and Independent Research Project of State Key Laboratory of Trauma and Chemical Poisoning (SKLYQ202101).

## Additional information

### Funding

| Funder | Grant reference number | Author |
| --- | --- | --- |
| National Key Research and Development Program of China | 2020YFA0908500 | Sheng Ye Cheng Zhu |
| National Natural Science Foundation of China | 22007071 | Cheng Zhu |
| National Natural Science Foundation of China | 31971127 | Sheng Ye |
| Natural Science Basic Research Program Key Projects of Shaanxi Province | 2024JC-ZDXM-42 | Rongrong Liu |
| National Natural Science Foundation of China | 82272330 | Xingan Wu |
| National Natural Science Foundation of China | 32100122 | Yangang Liu |
| National Natural Science Foundation of China | 62322215 | Fei Guo |
| Shanghai Sailing Program | 21YF1457800 | Yangang Liu |
| Independent Research Project of State of Key Laboratory of Trauma and Chemical Poisoning | SKLYQ202101 | Cheng Wang |

The funders had no role in study design, data collection and interpretation, or the decision to submit the work for publication.

### Author contributions

Ruihan Dong, Conceptualization, Data curation, Software, Formal analysis, Investigation, Visualization, Methodology, Writing – original draft, Writing – review and editing; Rongrong Liu, Resources, Formal analysis, Funding acquisition, Validation, Investigation, Visualization, Methodology; Ziyu Liu, Resources, Formal analysis, Validation, Investigation, Visualization, Methodology; Yangang Liu, Resources, Funding acquisition, Validation, Visualization, Methodology; Gaomei Zhao, Resources, Validation, Investigation, Visualization, Methodology; Honglei Li, Conceptualization, Investigation, Project administration; Shiyuan Hou, Xiaohan Ma, Huarui Kang, Jing Liu, Validation, Investigation; Fei Guo, Resources, Software, Funding acquisition; Ping Zhao, Junping Wang, Resources, Supervision, Validation; Cheng Wang, Resources, Funding acquisition, Validation, Investigation, Visualization, Methodology, Writing – review and editing; Xingan Wu, Resources, Supervision, Funding acquisition;

Sheng Ye, Resources, Supervision, Funding acquisition, Project administration; Cheng Zhu, Conceptualization, Data curation, Formal analysis, Supervision, Funding acquisition, Investigation, Methodology, Writing – original draft, Project administration, Writing – review and editing

## Author ORCIDs
Ruihan Dong ⓘ https://orcid.org/0009-0001-5862-8410
Rongrong Liu ⓘ https://orcid.org/0000-0002-9725-3463
Cheng Wang ⓘ https://orcid.org/0000-0002-6690-6433
Cheng Zhu ⓘ https://orcid.org/0000-0003-0260-6287

## Ethics
Mice were cared for and treated in accordance with the NIH guidelines for the care and use of laboratory animals (NIH Publication No. 85e23 Rev. 1985). Animal experiments were conducted with approval from the Animal Experimental Ethics Committee of the Army Medical University (AMUWEC20223290).

Reviewer #1 (Public review): https://doi.org/10.7554/eLife.97330.3.sa1
Reviewer #2 (Public review): https://doi.org/10.7554/eLife.97330.3.sa2
Reviewer #3 (Public review): https://doi.org/10.7554/eLife.97330.3.sa3
Author response https://doi.org/10.7554/eLife.97330.3.sa4

---

# Additional files

## Supplementary files
Supplementary file 1. Supplementary tables. (**a**) Ablation study results of AMPredictor. Best metrics are marked in bold and second-best values are underlined. (**b**) Information about five used antiviral classifiers. (**c**) Sequences and novelty (BLAST E-value) of validated peptides. (**d**) The $EC_{50}$ (µM) of AMPs inhibiting four enveloped viruses. (**e**) The selectivity index (SI) of three AMPs inhibiting four enveloped viruses. (**f**) Reported and predicted MICs of some recently mined AMPs. (**g**) Minimal inhibitory concentrations (µM) of P076-NH$_2$ and P076. (**h**) The qRT-PCR primers. (**i**) System details of molecular dynamics simulations. (**j**) Binding free energy between peptides and the lipid bilayers.

Supplementary file 2. Information of the generated 104 AMP sequences.

MDAR checklist

## Data availability
All the python codes of the AMP design model are available at https://github.com/ruihan-dong/GAN-for-AMP-Design, copy archived at *Dong, 2024a* and https://github.com/ruihan-dong/AMPredictor, copy archived at *Dong, 2024b*. An online version of AMPredictor can be accessed via https://huggingface.co/spaces/ruihan-dong/AMPredictor. The dataset to run AMPredictor is archived at https://doi.org/10.5281/zenodo.14955068.

The following dataset was generated:

| Author(s) | Year | Dataset title | Dataset URL | Database and Identifier |
|---|---|---|---|---|
| Dong R | 2022 | AMPredictor | https://doi.org/10.5281/zenodo.14955068 | Zenodo, 10.5281/zenodo.14955068 |

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
