## [Editor Report · eLife Assessment]

This study presents a **useful** pipeline for de novo design of antimicrobial peptides active both against bacteria and viruses. The method is based on deep learning, using a GAN generator and a regression tasked to predict antimicrobial activity. The experimental evidence supporting the conclusions is **solid**, with 24 validated peptides, although some additional justifications of the computational strategy would be a plus. This work will be of interest to the community working on machine learning for biomedical applications and specifically on antimicrobial peptides.

---

## [Referee Report · Reviewer #1 (Public review)]

This manuscript presents a pipeline incorporating a deep generative model and peptide property predictors for the de novo design of peptide sequences with dual antimicrobial/antiviral functions. The authors synthesized and experimentally validated three peptides designed by the pipeline, demonstrating antimicrobial and antiviral activities, with one leading peptide exhibiting antimicrobial efficacy in animal models.

Overall, the authors have addressed each major comment through new experiments, particularly by validating 24 peptides, clarifying alignment methods, and demonstrating sequence novelty. These additions have strengthened the manuscript. To further refine the work, it would be helpful to briefly describe any steps taken to mitigate GAN pathologies (such as mode collapse), provide a short rationale for the use of five AVP classifiers and how they complement each other, and clearly present the expanded experimental data (including MIC values and antiviral results) in the main text. Finally, the authors should also compare their approach with recently described deep-learning-enabled antibiotic discovery methods.

---

## [Referee Report · Reviewer #2 (Public review)]

Summary:

This study marks a noteworthy advance in the targeted design of AMPs, leveraging a pioneering deep learning framework to generate potent bifunctional peptides with specificity against both bacteria and viruses. The introduction of a GAN for generation and a GCN-based AMPredictor for MIC predictions is methodologically robust and a major stride in computational biology. Experimental validation in vitro and in animal models, notably with the highly potent P076 against a multidrug-resistant bacterium and P002's broad-spectrum viral inhibition, underpins the strength of their evidence. The findings are significant, showcasing not just promising therapeutic candidates, but also demonstrating a replicable means to rapidly develop new antimicrobials against the threat of drug-resistant pathogens.

Strengths:

The de novo AMP design framework combines a generative adversarial network (GAN) with an AMP predictor (AMPredictor), which is a novel approach in the field. The integration of deep generative models and graph-encoding activity regressors for discovering bifunctional AMPs is cutting-edge and addresses the need for new antimicrobial agents against drug-resistant pathogens. The in vitro and in vivo experimental validations of the AMPs provide strong evidence to support the computational predictions. The successful inhibition of a spectrum of pathogens in vitro and in animal models gives credibility to the claims. The discovery of effective peptides, such as P076, which demonstrates potent bactericidal activity against multidrug-resistant A. baumannii with low cytotoxicity, is noteworthy. This could have far-reaching implications for addressing antibiotic resistance. The demonstrated activity of the peptides against both bacterial and viral pathogens suggests that the discovered AMPs have a wide therapeutic potential and could be effective against a range of pathogens.

Comments on revisions: I have no further comments on revisions.

---

## [Referee Report · Reviewer #3 (Public review)]

Summary:

Dong et al. described a deep learning-based framework of antimicrobial (AMP) generator and regressor to design and rank de novo antimicrobial peptides (AMPs). For generated AMPs, they predicted their minimum inhibitory concentration (MIC) using a model that combines the Morgan fingerprint, contact map and ESM language model. For their selected AMPs based on predicted MIC, they also use a combination of antiviral peptide (AVP) prediction models to select AMPs with potential antiviral activity. They experimentally validated 3 candidates for antimicrobial activity against S. aureus, A. baumannii, *E. coli*, and P. aeruginosa, and their toxicity on mouse blood and three human cell lines. The authors select their most promising AMP (P076) for in vivo experiments in A. baumannii-infected mice. They finally test the antiviral activity of their 3 AMPs against viruses.

Strengths:

- The development of de novo antimicrobial peptides (AMPs) with the novelty of being bifunctional (antimicrobial and antiviral activity).

- Novel, combined approach to AMP activity prediction from their amino acid sequence.

Weaknesses:

- I missed the justification for combined antiviral and antibacterial activities. As the authors responded, less than 10% of the training data has antiviral activity. Therefore, I do not understand how the high percentage of antiviral activities was achieved. Especially reading that the antiviral filtering did not have an influence on the number of antiviral peptides obtained.

- I had difficulty in reading the story because of the use of acronyms without referring to their full name for the first time, and incomplete information annotation in figures and captions.

---

## [Author Response]

The following is the authors’ response to the original reviews.

**Public Reviews:**

**Reviewer #1 (Public Review):**
This manuscript presents a pipeline incorporating a deep generative model and peptide property predictors for the de novo design of peptide sequences with dual antimicrobial/antiviral functions. The authors synthesized and experimentally validated three peptides designed by the pipeline, demonstrating antimicrobial and antiviral activities, with one leading peptide exhibiting antimicrobial efficacy in animal models. However, the manuscript as it stands, has several major limitations on the computational side.

Thanks for your comments.

Major issues:(1) The choice of GAN as the generative model. There are multiple deep generative frameworks (e.g., language models, VAEs, and diffusion models), and GANs are known for their training difficulty and mode collapse. Could the authors elaborate on the specific rationale behind choosing GANs for this task?

We thank the reviewer for his/her concern on GAN models. We agree that there are some limitations of GAN itself such as its training difficulty, but we cannot deny its potential in generating biological sequences, especially in AMP generation. GAN and VAE are the two most commonly used generative models in the field of AMP design (Curr Opin Struct Biol 2023, 83:102733). AMPGAN (J Chem Inf Model, 2021, 61, 2198-2207.), Multi-CGAN (J Chem Inf Model 2024, 64, 1, 316–326), PepGAN (ACS Omega, 2020, 5, 22847-22851) and others have verified its application ability on peptide design. Moreover, PandoraGAN (Sn Comput Sci 2023, 4, 607) is one of the few works on AVP generation which is also based on GAN architecture. GAN updates the generator weights on the backpropagation from the discriminator directly rather than manually defined complicated loss function, which alleviates the reliance on input data. Our current results demonstrated that the trained GAN generator could produce novel sequences that featured high antimicrobial activity, both validated *in silico* and in vitro.

(2) The pipeline is supposed to generate peptides showing dual properties. Why were antiviral peptides not used to train the GAN? Would adding antiviral peptides into the training lead to a higher chance of getting antiviral generations?

A major mechanism of antimicrobial peptides is to disrupt cell membranes. Thus, some antimicrobial peptides are reported with broad-spectrum antibacterial and antiviral activities, since the virus shares a membrane structure with bacteria, especially the enveloped viruses. In APD3 database, 244 of 3940 AMPs are labeled with antiviral activities. In constrast, most reported antiviral peptides inhibit the viruses by binding to specific targets (proteins and nucleic acids) related to viral proliferation so that they may not have antibacterial effects. Therefore, we trained the GAN with the AMP dataset. We chose this AMP dataset mainly for AMPredictor (with detailed logMIC label against *E. coli*) and then used the same dataset to train a GAN for simplification.

In the revised manuscript, we also tested adding available antiviral peptides from AVPdb to train the GAN model. The number of AVPs is 1,788 after removing overlaps with used AMP dataset. The GAN architecture and hyperparameters remain the same. After generating a batch of sequences with this trained generator, we scored them by AMPredictor and filtered them with five AVP classifiers. As expected, the predicted MIC values shifted to higher performance with 17 sequences < 5 μM and 39 sequences < 10 uM, and previous numbers are 26 and 42 in the manuscript. Among 39 sequences < 10 μM, 13 passed all five AVP classifiers and 17 passed four (33.3% and 43.6%, respectively). Previous ratios are 40.5% and 35.7% (17 and 15 out of 42). Two generators perform roughly the same for generating AVPs (76.9% vs. 76.1%) as evaluated by our rules (4 or more positives), but the generator trained solely with AMPs provided more AVPs with higher possibility (5 positives).

We also experimentally tested dozens of generated peptides from two versions of generators (v1 for training solely on AMPs, v2 for training with AVPs, Figure 2 in revised manuscript). The ‘antiviral’ feature of a peptide was checked when significant inhibition was observed in immunofluorescence assays against HSV-1 at the concentration of 10 µM. Six and seven antiviral peptides were found out of 12 tested peptides from generators v1 and v2, respectively. Therefore, the success rates for two versions of generators are about 60% (including three reported peptides in the original manuscript) and show no significant difference.

(3) For the antimicrobial peptide predictor, where were the contact maps of peptides sourced from?

The contact maps of AMPs were predicted from ESM, which were obtained at the same time when obtaining the ESM embeddings (Methods section, Page 24, Line 538: Pretrained language model esm1b_t33_650M_UR50S was used to provide the embeddings and the contact maps.)

(4) Morgan fingerprint can be used to generate amino acid features. Would it be better to concatenate ESM features with amino acid-level fingerprints and use them as node features of GNN?

We thank the reviewer for this suggestion. We test using ESM and fingerprint (FP) features on graph nodes and the result is shown in Author response table 1. AMPredictor (ESM on nodes, FP after GNN) still performed slightly better than concatenating FP on node features on four regression metrics.

**Author response table 1. sa4table1:** Results of AMPredictor with fingerprint on nodes.

Model	RMSE	MSE	Pearson	CI
AMPredictor (ESM on nodes, FP after GNN)	0.5348	0.2860	0.7072	0.7294
ESM + FP on nodes	0.5356	0.2869	0.7012	0.7265

(5) Although the number of labeled antiviral peptides may be limited, the input features (ESM embeddings) should be predictive enough when coupled with shallow neural networks. Have the authors tried simple GNNs on antiviral prediction and compared the prediction performance to those of existing tools?

We thank the reviewer for his/her suggestion on AVP predictions. We haven’t tried it. An important reason is that we focused on developing regressors instead of binary classifiers. Currently available AVP data with numerical labels did not support training a reliable regressor, for their limited amount as well as heterogenous virus target and experimental assay. Therefore, we decided to use reported AVP classifiers as an additional filter following AMPredictor. Since only using one classifier may lead to bias, we chose five AVP classifiers as ensemble votes.

(6) Instead of using global alignment to get match scores, the authors should use local alignment.

We calculated the match scores by global alignment methods referred to AMPGAN v2 (J Chem Inf Model 2021, 61, 2198−2207), CLaSS (Nat Biomed Eng 2021 5, 613–623), and AMPTrans-lstm (Comput Struct Biotechnol J 2022, 21, 463-471), to check the similarity between the generated sequences and any sequences in the training set. In addition, we also used local alignment to check the novelty of peptides (regarding the next question).

(7) How novel are the validated peptides? The authors should run a sequence alignment to get the most similar known AMP for each validated peptide, and analyze whether they are similar.

We have listed the most similar AMP segments to our generated peptides from the training set and DRAMP database (28,233 sequences after filtering out those containing irregular characters). BLAST parameters were set as CLaSS (Nat Biomed Eng 2021 5, 613–623) for short peptides. The lowest Evalue of P001 aligned with the training set is 1.2, and no hits were found for P001 with DRAMP. Two E-values of P002 are 1.4 and 0.46. P076 had no hits in the training set and got a high E-value of 7.0 with DRAMP. Detailed alignments are shown below. This result indicates that our three validated AMPs are novel.

Since we generated more sequences using two versions of generator for validation, we also checked the BLAST E-value of these validated peptides. The results are listed in Table S3. All sequences obtained E-values > 0.1 and some of them had no hits when aligned with the training set or the DRAMP database.

**Author response image 1. sa4fig1:** Alignments of three validated peptides.

(8) Only three peptides were synthesized and experimentally validated. This is too few and unacceptable in this field currently. The standard is to synthesize and characterize several dozens of peptides at the very least to have a robust study.

We thank the reviewer for the suggestion and promoted our models to generate >10 times more peptides in the revised manuscript. We have synthesized and tested more peptides in vitro and added these results in the revised manuscript (Figure 2). From two versions of generators (trained with or without AVPs), we selected 24 peptides in total for antibacterial and antiviral validations. All 24 peptides showed antibacterial activity towards at least bacterial strain, and 13 peptides were screened out through the quick antiviral test. This result indicates the capability of our design method for bifunctional AMPs with a notable success rate (60%).

**Reviewer #2 (Public Review):**
Summary:This study marks a noteworthy advance in the targeted design of AMPs, leveraging a pioneering deeplearning framework to generate potent bifunctional peptides with specificity against both bacteria and viruses. The introduction of a GAN for generation and a GCN-based AMPredictor for MIC predictions is methodologically robust and a major stride in computational biology. Experimental validation in vitro and in animal models, notably with the highly potent P076 against a multidrug-resistant bacterium and P002's broad-spectrum viral inhibition, underpins the strength of their evidence. The findings are significant, showcasing not just promising therapeutic candidates, but also demonstrating a replicable means to rapidly develop new antimicrobials against the threat of drug-resistant pathogens.Strengths:The de novo AMP design framework combines a generative adversarial network (GAN) with an AMP predictor (AMPredictor), which is a novel approach in the field. The integration of deep generative models and graph-encoding activity regressors for discovering bifunctional AMPs is cutting-edge and addresses the need for new antimicrobial agents against drug-resistant pathogens. The in vitro and in vivo experimental validations of the AMPs provide strong evidence to support the computational predictions. The successful inhibition of a spectrum of pathogens in vitro and in animal models gives credibility to the claims. The discovery of effective peptides, such as P076, which demonstrates potent bactericidal activity against multidrug-resistant A. baumannii with low cytotoxicity, is noteworthy. This could have far-reaching implications for addressing antibiotic resistance. The demonstrated activity of the peptides against both bacterial and viral pathogens suggests that the discovered AMPs have a wide therapeutic potential and could be effective against a range of pathogens.

We thank the reviewer for the comments.

**Reviewer #3 (Public Review):**
Summary:Dong et al. described a deep learning-based framework of antimicrobial (AMP) generator and regressor to design and rank de novo antimicrobial peptides (AMPs). For generated AMPs, they predicted their minimum inhibitory concentration (MIC) using a model that combines the Morgan fingerprint, contact map, and ESM language model. For their selected AMPs based on predicted MIC, they also use a combination of antiviral peptide (AVP) prediction models to select AMPs with potential antiviral activity. They experimentally validated 3 candidates for antimicrobial activity against S. aureus, A. baumannii, *E. coli*, and P. aeruginosa, and their toxicity on mouse blood and three human cell lines. The authors select their most promising AMP (P076) for in vivo experiments in A. baumannii-infected mice. They finally test the antiviral activity of their 3 AMPs against viruses.Strengths:-The development of de novo antimicrobial peptides (AMPs) with the novelty of being bifunctional (antimicrobial and antiviral activity).-Novel, combined approach to AMP activity prediction from their amino acid sequence.Weaknesses:(1) I missed justification on why training AMPs without information of their antiviral activity would generate AMPs that could also have antiviral activity with such high frequency (32 out of 104).

Thanks for your inquiry. A major mechanism of antimicrobial peptides is to disrupt cell membranes. Thus, some antimicrobial peptides are reported with broad-spectrum antibacterial and antiviral activities, since the virus shares a membrane structure with bacteria, especially the enveloped viruses. In APD3 database, 244 of 3940 AMPs are labeled with antiviral activities. However, several reported antiviral peptides inhibit the viruses by binding to specific targets (proteins and nucleic acids) related to viral proliferation so that they may not have antibacterial effects. Therefore, we trained the GAN with the AMP dataset. We chose this AMP dataset mainly for AMPredictor (with detailed logMIC label against *E. coli*) and then used the same dataset to train a GAN for simplification. In addition, it’s not 32 antiviral candidates out of 104 but 32 out of 42 peptides with predicted MIC < 10 µM because we did the filtering process stepwise.

In revision, we also tested adding available antiviral peptides from AVPdb to train the GAN model (generator v2). The number of AVPs is 1,788 after removing overlaps with used AMP dataset. The GAN architecture and hyperparameters remain the same. We used generator v2 to obtain a batch of sequences and screened out bifunctional candidates following the same procedure. 30 out of 39 peptides with predicted MIC < 10 µM passed four or five AVP predictors. Therefore, two generators perform roughly the same for generating AVP candidates (76.9% vs. 76.1%).

(2) The justification for AMP predictor advantages over previous tools lacks rationale, comparison with previous tools (e.g., with the very successful AMP prediction approach described by Ma et al. 10.1038/s41587-022-01226-0), and proper referencing.

Thanks for your suggestion. Ma et al. proposed ensemble binary classification models to mine AMPs from metagenomes successfully. However, we concentrated on the development of regression models. As a regressor, AMPredictor predicts the specific logMIC value of the input sequences instead of giving a yes/no answer. Since the training settings and evaluation metrics are different for the classification and regression tasks, we could not compare AMPredictor with Ma et al. directly. Instead, we compared the performance of AMPredictor with some regression baseline models (Figure 1-figure supplement 2) and our model outperformed them.

(3) Experimental validation of three de novo AMPs is a very low number compared to recent similar studies.

Thanks for pointing out this shortcoming. We have synthesized and tested more peptides in vitro and added these results in the revised manuscript (Figure 2). From two versions of generators (trained with or without AVPs), we selected 24 peptides in total for antibacterial and antiviral validations. All 24 peptides showed antibacterial activity towards at least bacterial strain, and 13 peptides were screened out through the quick antiviral test. This result indicates the capability of our design method for bifunctional AMPs with a notable success rate (60%).

(4) I have concerns regarding the in vivo experiments including (i) the short period of reported survival compared to recent studies (0.1038/s41587-022-01226-0, 10.1016/j.chom.2023.07.001, 0.1038/s41551-022-00991-2) and (ii) although in Figure 2 f and g statistics have been provided, log scale y-axis would provide a better comparative representation of different conditions.

Thank you for your suggestions.

i) In current study, we monitored the survival of mice with peritoneal bacterial infection for 48 h.

Because abdominal bacterial infection can induce severe sepsis and cause mouse death within 40 h (Sci Adv 2019, 5(7), eaax1946), the 48 h is sufficient to evaluate the therapeutic efficacy of antimicrobial peptides (Nat Biotechnol 2019, 37(10), 1186-1197).

ii) In Figure 2f and 2g (3f and 3g in the revised manuscript), the y-axis has already been in log-scale and tick labels are marked in scientific notation.

(5) I had difficulty reading the story because of the use of acronyms without referring to their full name for the first time, and incomplete annotation in figures and captions.

Thank you for pointing this. We have checked the manuscript carefully and modified the figure captions during revision.

**Reviewer #2 (Recommendations For The Authors):**
(1) To validate the generalizability of the model, it would be prudent to include data on AMPs targeting a broader range of bacteria and viruses. This could help ensure that the peptides designed are not narrowly focused on *E. coli* but are effective against a more extensive set of pathogens.

Thanks for your suggestions. We just incorporated AMPs with *E. coli* activity labels since it is the most common strain among available AMP databases. As for a regressive model (AMPredictor), the fitting object should be defined concisely, which means limited targeting bacteria. Some other articles also focused on *E. coli* labels as well (Nat Commun 2023, 14, 7197; mSystems 2023, 8, e0034523).

We used the same processed dataset to train the GAN generator for simplification. Most reported AMPs have the potential to target various microbes. We have counted the antimicrobial labels of these peptides in our dataset, shown in Figure 1-figure supplement 1b. In addition to *E. coli*, some of the peptides target Grampositive *S. aureus*, fungus *C. albicans,* and other bacterial species as well. Our experimental validation also reveals the wide spectrum of designed peptides inhibiting Gram-negative, Gram-positive, drugresistant bacteria, and enveloped viruses. With the expansion of well-curated AMP databases, we expect to update the model with larger scale datasets in the near future.

(2) Conduct sensitivity analyses to understand how minor changes in the peptide sequences impact the model’s predictions. This will reduce the chances of overlooking potential AMP candidates due to the model’s inability to capture subtle changes.

Thank you for this valuable suggestion. We kept similar known peptide sequences in the training sets regarding that a single mutation may have an impact on their antimicrobial performances. We took P001 as an example to perform the sensitivity analysis by site saturation mutagenesis *in silico*. Author response image 2 represents the change of antimicrobial activity scores as predicted by AMPredictor. Since the predicted MIC of P001 is 0.949 µM (experimentally measured value is 0.80 µM), most single mutations lead to higher scores (i.e., worse performance), especially Asp (D) and Glu (E) residues with negative charges. The largest change value of single amino acid replacement is 25.51 (W6D). Although this value may not reflect the actual changes, it is enough to be distinguished when screening and ranking candidate sequences.

**Author response image 2. sa4fig2:** Site saturated mutagenesis of P001. Color shows the change of predicted MIC against *E. coli* as predicted by AMPredictor (lower score is better).

(3) Given the relatively short length of the peptides, typically ranging from 10 to 20 residues, the authors might consider employing a fully-connected graph in the peptide’s graphical representation. This approach could potentially simplify the model without sacrificing the descriptive power due to the limited size of the peptides.

Thanks for your suggestions. We tested fully-connected graph edge encodings and the results on the test set were shown in Author response table 2 below. We found that AMPredictor with peptide contact map still performed better on Pearson correlation coefficient and CI, while using fully-connected graphs reached a slightly improved RMSE and MSE. Nonetheless, using fully-connected graph demands about 10time memory and more computational costs when processing more complicated message-passing. Therefore, the involvement of structural information is still a preferred choice.

**Author response table 2. sa4table2:** Results of AMPredictor with different graph edge encodings.

Graph edge encoding	RMSE	MSE	Pearson	CI
Peptide Contact Map	0.5348	0.2860	0.7072	0.7294
Fully-connected	0.5319	0.2828	0.6946	0.7250

(4) Upon reviewing Table S1, it is apparent that the application of ESM embeddings alone achieves commendable prediction accuracy. It would be intriguing to investigate whether the adoption of the more recent ESM models-specifically the second-generation ESM2 t36_3B, t48_15B, and t33_650Mcould enhance model performance beyond that observed using the esm1b_t33_650M_UR50S model described in the manuscript.

Thanks for your suggestions. Here, we included various ESM2 models’ outputs as our node features and presented the results in Author response table 3. Notably, the dimensions of esm2_t36_3B and esm2_t48_15B are 2560 and 5120, respectively, while both esm2_t33_650M and esm1b_t33_650M are 1280 dimensions.

Interestingly, we found that larger models don’t lead to improved performance. ESM-1b version still holds the best metrics in RMSE, MSE, and Pearson correlation coefficient. This indicates that the choice of pretrained model versions depended on specific downstream tasks.

**Author response table 3. sa4table3:** Results of AMPredictor with different ESM versions.

ESM embedding	RMSE	MSE	Pearson	CI
esm1b_t33_650M_UR50S	0.5348	0.2860	0.7072	0.7294
esm2_t33_650M_UR50D	0.5425	0.2943	0.6944	0.7563
esm2_t36_3B_UR50D	0.5581	0.3115	0.6774	0.7494
esm2_t48_15B_UR50D	0.5622	0.3160	0.6503	0.7093

(5) It may be pertinent to reevaluate the use of the MM-PBSA approach within the scope of this study. Typically, MM-PBSA is utilized to estimate the free energy differences between the bound and unbound states of solvated molecules. The application of MM-PBSA is to calculate binding energies between proteins and membranes is unconventional and infrequently documented in the literature. Therefore, it is recommended that the authors consider omitting this portion of the manuscript, or provide a robust justification for its inclusion and application in this context.

Thanks for your comments on MM/PBSA methods. There have been several literatures using this approach to calculate peptide-membrane binding free energy (Langmuir 2016, 32, 1782-1790; J Cell Biochem 2018, 119, 9205-9216; J Chem Inf Model 2019, 59, 3262-3276; Molecular Therapy Oncolytics 2019, 16, 7-19; Microbiology Spectrum 2023, 11, e0320622; J Chem Inf Model 2023, 63, 5823-5833) and we referred to their settings, such as the dielectric constant. All of these works built similar all-atom systems including cationic antimicrobial peptides and membrane bilayers, and utilized MM/PBSA method to describe the absorption process of the peptide from an unbound initial state. The order of magnitude of our calculation results is consistent with other reported works. Additionally, computational results may provide supporting evidence and we discussed that this quantitative energy calculation should be considered along with other observed metrics.

**Reviewer #3 (Recommendations For The Authors):**
The weaknesses I mentioned in the Public Review may be addressed by improving the writing and presentation and corrections to the text and figures.

Thanks for your suggestion. We have carefully checked and improved the presentation of text and figures in the revised manuscript.